# Silver-Coated Silica Nanoparticles Modified with MPS: Potential Antimicrobial Biomaterials Applied in Glaze and Soft Reliner

**DOI:** 10.3390/polym14204306

**Published:** 2022-10-13

**Authors:** Natália Rivoli Rossi, Beatriz Rossi Canuto de Menezes, Aline da Graça Sampaio, Diego Morais da Silva, Cristiane Yumi Koga-Ito, Gilmar Patrocínio Thim, Tarcisio José de Arruda Paes-Junior

**Affiliations:** 1Department of Dental Materials and Prosthodontics, Institute of Science and Technology, São Paulo State University (UNESP), São José dos Campos 12245-000, Brazil; 2Plasma and Process Laboratory, Aeronautical Technology Institute (ITA), São José dos Campos 12228-970, Brazil; 3Department of Environmental Engineering and Sciences Applied to Oral Health Graduate Program, Institute of Science and Technology, São Paulo State University (UNESP), São José dos Campos 12245-000, Brazil

**Keywords:** nanoparticles, silver, polymers, denture liners, biofilms

## Abstract

Soft reliner and glaze are materials used over full or partial dental prosthesis to prevent excessive pressure on the supporting tissues. They are also indicated as supportive treatment for dental stomatitis, especially when modified by the addition of medications. The objective of the work was to evaluate the antimicrobial effect of silver-coated silica nanoparticles in a glaze and a soft reliner. The nanoparticles were synthesized, characterized, and tested by minimum inhibitory concentration (MIC) for *C. albicans* SC5314. Then, the nanoparticles were incorporated to a glaze and a soft reliner, which were called nanocomposites. Then, the nanocomposites were divided into six groups (*n* = 12): CG: glaze/reliner; CR: reliner; G1: glaze + 1% nanoparticles/reliner; G2: glaze + 2.5% nanoparticles/reliner; R1: reliner + 1%; R2: reliner + 2.5%. The nanocomposites were characterized by a goniometer and by a scanning electron microscope. The antibiofilm test was performed against *C. albicans* SC5314. According to the MIC results, the non-functionalized nanoparticles reduced fungal growth at 1000 μg/mL and the functionalized nanoparticles at 2000 μg/mL. The functionalized nanoparticle had a superior dispersion being selected for the antibiofilm test. There was a reduction of 64% in CFU/specimen count for the glaze, not statistically significant (*p* = 0.244). For the soft reliner, there was an increase in CFU/specimen with the presence of nanoparticles, still not statistically significant (*p* = 0.264). In conclusion, it is necessary to conduct new studies to increase the release of silver, thus improving nanoparticles’ antifungal potential.

## 1. Introduction

The most extensively used polymer in producing removable full or partial dentures is acrylic resin. Acrylic resin is known for its good aesthetics, low cost, easy handling, and proper biocompatibility [1]. An exothermic reaction during the acrylic resin polymerization process converts the monomer methyl methacrylate into polymer, resulting in a hard and brittle material. This material may have pores on its surface which can absorb small amounts of water when placed in a humid environment, being a favorable place for biofilm growth [2].

Denture liners are also used in treating patients who use removable full or partial dentures, being called tissue conditioners. These materials can be rigid or resilient and may or may not have a glaze to impermeabilize its surface. They are used on the surface of full dentures while tissue is healing [3]. In addition to protecting surgical wounds, they also condition the supporting mucosa damaged by the poor adaptation of prostheses, generating comfort for the patient. They are also indicated as supportive treatment for dental stomatitis, especially when modified by adding medications [4]. Furthermore, denture liners are also used to improve the fit of removable dentures and to avoid excessive pressure on the supporting tissues. They can be composed of polymers based on methylmethacrylate, or based on other polymers.

Biofilm can colonize the dental prosthesis surface, resulting in oral lesions, in particular prosthetic stomatitis [5,6]. Furthermore, microbial reservoirs can facilitate the formation of caries and the occurrence of periodontal diseases in the remaining teeth. Such conditions harm the longevity of dental rehabilitation and constitute a risk of opportunistic infections, reducing the quality of life and generating new expenses for patients. For older patients with compromised immune activity, an accumulation of bacteria can favor systemic diseases, which are among the dominant causes of death and morbidity [7,8].

There is no material available on the dental market which can inhibit microorganisms aggregating on the surface of acrylic resin. Thus, there is a need to create a new material with antimicrobial and antifungal properties whose objective is to form a protective layer capable of inhibiting bacterial and fungal proliferation on the surface of prostheses. In this context, several nanomaterials have been studied by combining biocompatible polymers and inorganic materials, such as SiO_2_, TiO_2_, ZnO, and silver, among others [9,10,11,12,13].

Silver nanoparticles (AgNPs) stand out as one of the most intensively investigated systems. They have unique properties, such as excellent electrical conductivity, catalytic activity, non-linear optical behavior, high stability and resistance to oxidation process, and antimicrobial effects, making them potential candidates for applications in different areas including sensors, textiles, electronics, catalysis, food packaging, and medical therapy [14,15,16]. The use of AgNPs expanded in the dental field because of their bactericidal and bacteriostatic activity against microorganisms such as fungi, bacteria, and viruses [17,18]. Chladek et al. [13] modified silicon soft line polymer with AgNPs (10 to 200 ppm of AgNPs) and evaluated the antifungal efficacy (AFE) of the developed nanocomposites. The results showed AFE varying between 16.4% (10 ppm) to 52.2% (200 pm) for the examined samples. According to the authors, increasing the amount of AgNPs increased the antifungal behavior.

However, the work of Habibzadeh et al. [19] showed that the addition of AgNPs to silicone soft liner reduced its tensile bond strength. The authors also reported that the release of AgNPs into the oral cavity may increase its toxicity, needing a controlled release to maintain the optimal balance between the biocompatibility and antifungal activity of the nanocomposite. The cited works showed the potential of AgNPs to be used as antibacterial and antifungal materials; however, further studies should be carried out in order to control AgNP toxicity and its effects on polymeric mechanical properties.

In this context, the use of SiO_2_ nanoparticles have been showing great potential to be used as carrying material for antimicrobial particles. Metal oxides are commonly used due to their stability, biocompatibility, and for being essential minerals for human health [9]. Gad et al. [10] incorporated SiO_2_ nanoparticles into a soft relining denture material to reduce *Candida albicans* adhesion. According to the results, the addition of 0.25% and 0.5% of nano-SiO_2_ significantly reduced adhesion of *C. albicans* fungus onto acrylic soft liner. However, incorporating 1.0% of nano-SiO_2_ resulted in higher *C. albicans* counts when compared to the 0.25% and 0.5% samples. The authors suggested that higher nano-SiO_2_ concentration can lead to their agglomeration in the polymeric matrix.

The combination of antifungal and antibacterial properties of AgNPs with the biocompatibility and stability of SiO_2_ nanoparticles has great potential. Loading AgNPs in SiO_2_ surfaces has showed great potential to regulate the solubility and toxicity of Ag^+^ free ions by controlling their release [20]. SiO_2_ nanoparticles enable loading large amounts of antimicrobial materials and the slow release of bactericidal agents for a prolonged period, being potentially used as carriers and a core for antimicrobial agents [21]. Many works have been developing techniques to use silver as nanomaterials for biomedical applications by improving its dispersibility, stability, and prolonging the release time. El-Nour et al. studied a methodology to control the release of Ag^+^ ions by incorporating SiO_2_ as a nucleus which acts as a silver support [15]. SiO_2_ is a paramount material for immobilizing nanoparticles on its surface due to its high chemical and thermal stability, chemical inertness, large surface area, and good compatibility with other materials [22,23]. Le et al. [20] evaluated incorporating SiO_2_-Ag hybrid nanofillers at concentrations of 0.5 wt% to 4.0 wt% on water-based acrylic coating. The authors studied the effect of hybrid nanofillers on the abrasion resistance, thermal stability, and antibacterial activity against *Escherichia coli*. According to the results, incorporating 2.0% of SiO_2_-Ag nanofillers simultaneously improves the coating’s mechanical properties, thermal resistance, and antibacterial activity. However, a higher amount of the nanofiller in the matrix (4.0 wt%) showed particle agglomeration, as shown by scanning electron microscopy images, reducing the adhesion and abrasion resistance.

As shown, SiO_2_ nanoparticles have great potential to be used as carriers and cores for antimicrobial agents, such as AgNPs. However, many works have indicated the tendency for agglomeration and reduction of compatibility of these nanomaterials with organic matrices, especially when the amount of nanofiller is superior to 2.0 wt%. The modification of the SiO_2_-AgNPs nanofiller surfaces is an alternative to improve their interaction and dispersion in the polymeric matrix. γ–methacryloxy propyl trimethoxy silane (MPS) is an important silane which is widely used as a functionalization agent to promote interfacial interactions between metal oxide nanoparticles and a polymeric matrix, such as dental acrylic resins. The silanization mechanism occurs by M−O−Si (M = metal) bonds between MeO-NPs and silane coupling agents [24].

Several works have shown improvements in silanization of nanofillers used in acrylic-based matrices for dental applications. Menezes et al. [24] modified the surface of silver vanadate nanorods (AgVO_3_) using MPS to enhance the dispersion and interaction of this filler with polymethyl methacrylate (PMMA). The authors showed that nanocomposites produced with MPS-modified AgVO_3_ had higher Shore-D hardness and impact strength values than a nanocomposite with pristine AgVO_3_ due to the improved dispersion and interaction.

This work reports a pioneer silver-coated SiO_2_ nanoparticle surface modification through silanization reaction using MPS and its incorporation into dental acrylic matrix. Glaze and soft reliner were used as the organic matrix due to their precise indication to reduce inflammation and distribute a balanced load on the oral tissues during dental repairs. A complete characterization of the MPS-silanized silver-coated SiO_2_ nanoparticles was performed. Then, an unmodified and modified nanofiller were incorporated into glaze and soft reliner polymers. The effect of the nanofillers in the matrix and antifungal activity was evaluated. Therefore, the aim of this study was to synthesize SiO_2_-Ag-MPS nanoparticles, to characterize them, to test the minimum inhibitory concentration against *Candida albicans*, and to perform the anti-biofilm test against *Candida albicans.*

## 2. Materials and Methods

### 2.1. Synthesis of SiO_2_ Nanoparticles

The SiO_2_ nanoparticles were produced using the Stöber method [25], which consists of producing monodisperse suspension of silica spheres in the colloidal size range through the hydrolysis and controlled condensation of tetraethylorthosilicate (TEOS, 98.0%, Si(OCH_3_)_4_, Sigma) in an alcoholic medium.

For this purpose, two different solutions were prepared separately: (1) “Solution A”: 9 mL of TEOS and 55 mL of ethyl alcohol (98.8%, C_2_H_6_O, Synth); and (2) “Solution B”: 30 mL of deionized water, 55 mL of ethyl alcohol, and 5 mL of ammonium hydroxide (24.0%, NH_4_OH, Synth). “Solution A” and “Solution B” were separately kept in magnetic stirring until homogenization. Then, “Solution B” was totally poured into “Solution A” and the mixture was kept under constant stirring for one hour until the condensation reaction was complete. After this process, the precipitate obtained was washed and centrifuged. Then, the material was dried in an oven at 60 °C for 12 h. The sample was labeled as SiO_2_.

### 2.2. Silver-Coating of SiO_2_ Nanoparticles

The SiO_2_ nanoparticles were coated with silver following the Nischala procedure [18]. First, 50 mL of the SiO_2_ suspension was mixed with a solution containing 12.5 mL of deionized water, 1.70 g of silver nitrate (99.9%, AgNO_3_, Neon) (10 mmol), and 2 g of anhydrous glucose (97.0%, C_6_H_12_O_6_, Neon). The mixture was subsequently stirred on a magnetic stirrer for 30 min at room temperature. Then, 25 mL of a sodium carbonate solution (25 mL of deionized water + 2 g of anhydrous sodium carbonate (99.5%, Na_2_CO_3_, Synth)) was added dropwise to the mixture and kept under stirring for a further one hour. After this process, the precipitate obtained was washed and centrifuged. Then, the material was dried in an oven at 60 °C for 12 h. The sample was labeled as SiO_2_-Ag.

### 2.3. Functionalization of SiO_2_-Ag Nanoparticles Using MPS

The functionalization of the SiO_2_-Ag nanoparticles was performed using γ-methacryloxy propyltri methoxy silane (MPS, 98.0%, H_2_C=C(CH_3_)CO_2_(CH_2_)_3_Si(OCH_3_)_3_, Sigma). First, 200 mL of deionized water and 2 g of nanoparticles were mixed and kept in an ultrasound bath (Bransson 2210, power 225 W) for 15 min. Then, 6.8 mL of MPS was added, which was kept under magnetic stirring at room temperature until homogenization. Finally, 3.4 mL of 1 M glacial acetic acid (100.0%, CH_3_COOH, Synth) was added dropwise. The mixture was kept under magnetic stirring at 450 rpm at room temperature for three hours. Then, the solution was centrifuged and washed with deionized water and alcohol several times. Finally, the obtained powder was kept in an oven at 60 °C for 12 h. The sample was labeled as SiO_2_-Ag-MPS. The full schematic diagram of the synthesis is displayed in Figure 1.

### 2.4. Microstructural Characterization of Nanoparticles

Scanning electron microscopy with energy dispersive X-ray spectroscopy (SEM-EDX) and field emission scanning electron microscopy (FE-SEM) were performed using Tescan Vega 3 with an accelerating voltage of 20 kV and using a Jeol JSM-5310 with accelerating voltage of 30 kV, respectively. For SEM analysis, SiO_2_, SiO_2_-Ag, and SiO_2_-Ag-MPS powders were placed on conductive carbon tape. X-ray powder diffraction (XRD) was performed using a PANalytical Philips X’Pert device with Cukα radiation operating at 45 kV and 40 mA in the range of 10° < 2θ < 90°, Δθ = 0.02°, and time of 30 s per Δθ. Fourier-transformed infrared spectroscopy (FT-IR) was performed in a PerkinElmer Spectrum One spectrophotometer using the attenuated total reflectance (ATR) technique, in the 2000–500 cm^−1^ region, with a resolution of 4 cm^−1^ and 16 scans.

### 2.5. Minimum Inhibitory Concentration of SiO_2_-Ag MPS Nanoparticles Solution for Candida albicans

The minimum inhibitory concentration (MIC) assay was performed for SiO_2_, SiO_2_-Ag, and SiO_2_-Ag-MPS using *Candida albicans* SC5314 fungal strain, according to the CLSI M27-A3 standard [26]. The MIC was determined based on the lowest antifungal concentration corresponding to 50% reduction compared to the control group.

A *C. albicans* strain stored at −80 °C in Sabouraud dextrose (SD) broth in 20% glycerol was cultivated on an SD agar plate for 24 h at 37 °C in aerobiosis. After cultivation, a standardized suspension of cells in sterile physiological solution (NaCl, 0.9%) was rinsed until reaching a concentration of 10^6^ cells/mL (wavelength (λ) 530 nm and optical density (OD) 0.138). An AJX-1600 spectrophotometer from AJMicronal standardized the inoculum. Then, the inoculum was cultured in an SD broth culture medium at a concentration of 10^3^ cells/mL.

The substances in this susceptibility test were cultured in an SD broth culture medium at a concentration of 4000 μg/mL for SiO_2_-Ag and 16,000 μg/mL for pure SiO_2_ and SiO_2_-Ag-MPS. Then, 200 μL was cultured to the first column with serial dilutions of ratio 2:1 executed in 100 μL of culture medium. Lastly, 100 μL of inoculum was cultured, so concentrations ranging from 2000 to 3.90 μg/mL for *SiO2-Ag* and 8000 to 15.6 μg/mL for pure SiO_2_ and SiO_2_-Ag-MPS were obtained. Positive and negative control were also included in the test. A growth control containing inoculum in a culture medium with an absence of the tested substance was a positive control. The negative control was only based on culture medium. Afterward, the plate incubations were performed in aerobiosis at 37 °C for 24 h with a triplicate test for each sample on three different days (*n* = 9).

The wells were subsequently subcultured in Petri dishes due to the dark coloration of the test samples (SiO_2_-Ag and SiO_2_-Ag-MPS), and incubated at 37 °C for 24 h. Then, 50% inhibition of growth determined the MIC values when compared to the growth control group.

### 2.6. Preparation of Nanocomposites with the Incorporation of SiO_2_-Ag-MPS to the Glaze and Soft Reliner

The SiO_2_-Ag-MPS nanoparticles were selected and aggregated into the glaze (soft comfort glaze, methyl methacrylate, Dencril) and the soft reliner (soft comfort soft reliner, ethyl methacrylate, Dencril), as its dispersion in the polymer was superior to the other nanoparticles. The concentrations regarding the nanocomposite mass were 1.0 wt% and 2.5 wt% [27]. Thus, three different experimental groups were defined for the glaze (Table 1) and three for the soft reliner (Table 2) (*n* = 12), with a negative control group (no nanoparticles), and two experimental groups of nanoparticles, varying the concentration concerning the polymer mass (1.0% and 2.5%).

Next, 36 cylindrical samples of 6 × 2 mm (diameter and thickness) were manufactured using soft reliner for three experimental groups in which the glaze was applied (CG, G1, and G2). Then, the materials of the two vials (powder and liquid) were mixed in the proportion recommended by the manufacturer (2:1) to prepare the soft reliner. First, 21 g of polymer powder (ethyl methacrylate) was agglutinated in 10 mL of liquid monomer in a glass pot with a lid. A syringe wielded the material during its plastic phase to avoid bubbles. Then, after the material settled inside metallic patterns of 6 × 2 mm (diameter × thickness), a glass plate was inserted over it, and a pressure of 350 g was applied on the set. The SiO_2_-Ag-MPS nanoparticles were subsequently dispersed in the liquid glaze (methyl methacrylate) in the G1 (1.0 wt%) and G2 (2.5 wt%) groups, and were kept under constant agitation for 15 min for complete incorporation. Glaze application was performed using a microbrush on the soft reliner, obtaining the CG, G1, and G2 nanocomposites.

Following these procedures, another 36 cylindrical soft reliner samples (6 × 2 mm) were produced and displayed in CR, R1 (1 wt%), and R2 (2.5 wt%) groups (reliner with nanoparticles). A modified monomer with nanoparticles was kept under constant agitation for 15 min to completely disperse the nanoparticles in the liquid for these groups. The fabrication of the specimens followed the same described procedure. After polymerization, the prepared discs were cleaned in an ultrasound with distilled water for 5 min, obtaining the CR, R1, and R2 nanocomposites.

### 2.7. Microstructural Characterization of Nanocomposites

The surface of the samples was analyzed by field emission scanning electron microscopy (FE-SEM) using a Jeol JSM-55310 device with accelerating voltage of 20 kV. The samples were previously recovered with a thin layer of gold deposition in a metallizer (EMITECH SC7620, Sputter Coater) for 120 s at 12 mA. The contact angle and surface energy analysis test were performed for all nanocomposites using a goniometer (Theta Lite model, Attension, Biolin Scientific, Espoo, Finland), under controlled temperature and connected to a computer with the One Attension software program. A drop of distilled water over the nanocomposite’s surface measured the contact angle after 10 s.

### 2.8. Microbiological Characterization of Nanocomposites—Evaluation of the Anti-Biofilm Activity of Candida albicans

Before the biofilm formation test, the nanocomposites underwent sterilization by exposure to ultraviolet radiation for 10 min on each face. After sterilization, the standard *C. albicans* SC 5314 strain was used. The *C. albicans* sample was seeded in Sabouraud dextrose agar and incubated for 24 h at 37 °C in aerobiosis. Then, a standardized suspension was prepared in a spectrophotometer (λ: 550; DO: 0.380), which contained a sterile physiological solution (0.9% NaCl, pH 7.0) in 106 cells/mL. The suspension was settled in wells of upright 96-well polystyrene plates. Then, 20 μL aliquots of the fungal inoculum, 200 μL of RPMI broth, and 2% glucose were mixed together and added to each well. The plates were incubated for 120 min at 37 °C in aerobiosis under agitation (80 rpm), so fungal cells could adhere to the specimens’ surface. After a pre-adherence period of 120 min, the samples were washed with a sterile saline solution to remove non-adherent cells. Then, the culture medium was added and incubated at 37 °C for 24 h. Afterward, the specimens settled in tubes containing 1 mL of saline solution. The tubes were vortexed for 60 s and then sonicated (two pulses of 30 s with a 20-s interval, amplitude 40, 15 W on ice) to recover the biofilm. The resulting suspension was serially diluted (up to 10–3) and plated on Sabouraud dextrose agar. The plates were incubated in a stove for 24 h at 37 °C in aerobiosis. Next, the viable biofilm cells were quantified by obtaining CFU/specimen values and then statistically compared between the test and growth control groups.

Two specimens from each group were selected and prepared under the conditions described above for analysis by scanning electron microscopy. The biofilms were subjected to the fixation process by 2.5% glutaraldehyde for 24 h. After rinsing in saline solution, different alcohol concentrations dehydrated the specimens (30%, 50%, 70%, 80%, 90% for 15 min, and 100% for 30 min). Then, the biofilms settled at room temperature for 24 h. Biofilms were metalized with a thin layer of gold (EMITECH SC7620, Sputter Coater) and analyzed by field emission scanning electron microscopy (FE-SEM) using a Jeol JSM-55310 with accelerating voltage of 20 kV.

### 2.9. Statistical Analysis

After compiling all the results, the normality and homoscedasticity test (Kolmogorov–Smirnov) assessed the existence of normal distribution. There was normal distribution for the contact angle, so means and standard deviations were calculated and submitted to one-way ANOVA and Tukey’s test (α = 0.05). There was non-normal distribution for the anti-biofilm test. We consequently used the Kruskal–Wallis test to evaluate the data.

## 3. Results

### 3.1. Synthesis of Nanoparticles

The synthesis of SiO_2_ was performed using the Stöber method. The simplified reactions of hydrolysis (1) and condensation (2) are [25]:(1)$$\;\mathrm{Si}{\left(\mathrm{OR}\right)}_{4}+4H_{2}O\;\to \mathrm{Si}{\left(\mathrm{OH}\right)}_{4}+4\mathrm{ROH}$$
(2)$$\mathrm{Si}{\left(\mathrm{OH}\right)}_{4}\to {\mathrm{SiO}}_{2}+4H_{2}O$$

In this reaction, TEOS is controlled and hydrolyzed (1) in a medium compounded by ethanol, producing Si(OH)_4_. Then, the dispersed silanol phase is condensed in a polymerization reaction (2) when the SiO_2_ nanoparticles are obtained.

A previous work regarding the Ag coating reaction on SiO_2_ nanoparticles reports that the addition of silver nitrate, glucose, and Na_2_CO_3_ to SiO_2_ suspension enables forming [Ag(NH_3_)_2_]^+^ ions which are electrostatically attracted to silanol groups. Then the ionic Ag is reduced to metallic Ag on SiO_2_ surfaces, obtaining the SiO_2_-Ag structure [18].

The reaction of MPS (RSi-OCH_3_) bonding to SiO_2_ was also already discussed in the literature. The hydroxyl groups on the SiO_2_ surface react with the OCH_3_ groups available in MPS molecules by acid hydrolysis [18,28]. Then, the obtained products are RSi-O-Si and methanol (3)—which are withdrawn in the washing process and not present in the final product (SiO_2_-Ag-MPS).
(3)$$\mathrm{RSi}-{\mathrm{OCH}}_{3}+\;\mathrm{Si}-\mathrm{OH}\;\to \;\mathrm{RSi}-O-\mathrm{Si}\;+{\;\mathrm{CH}}_{3}\mathrm{OH}$$

#### 3.1.1. Scanning Electron Microscopy—Field Emission Gun (FE-SEM)

Figure 2 presents the images obtained for SiO_2_, SiO_2_-Ag, and SiO_2_-Ag-MPS. Figure 2a shows spherical and uniform SiO_2_ nanoparticles with a smooth and homogeneous surface, and regular size. The average diameter ranged between 468 and 504 nm. After the silver coating process, smaller silver particles can be observed on SiO_2_ surfaces (Figure 2b). Those smaller particles (light gray) present a brighter color when compared to SiO_2_ matrix (dark gray), which corroborates the deposition of AgNPs, considering that heavier elements are brighter in the grayscale of SEM images [29]. The silver dispersed over silica was uniform and regular, and the diameter of the SiO_2_-Ag nanoparticles ranged between 423 and 510 nm. Figure 2c shows SiO_2_-Ag-MPS nanoparticles, which did not present a morphological difference regarding SiO_2_-Ag nanoparticles, maintaining the diameter between 418 to 502 nm. There were no regions of amorphous or unknown morphology, indicating that the functionalization was homogeneous, without morphological changes and clusters of functionalizing agents.

#### 3.1.2. Energy Dispersive Scanning Electron Microscopy (SEM-EDX)

SEM-EDX was used to analyze the chemical composition of the nanomaterials, specifically the percentage values for oxygen (O), silica (Si), and silver (Ag) present in each sample (Table 3). The carbon (C) concentration was not considered in the discussion since it is related to carbon conductive tape used on sample preparation and also with contaminations inside the SEM analysis chamber.

According to Table 3, the SiO_2_ sample showed only Si and O chemical components. After the silver-coating, SiO_2_-Ag showed a significant amount of Ag in the sample composition, over 33.2 wt%, indicating the incorporation of AgNPs on the SiO_2_ surfaces. After the functionalization process, there was an increase in Si, which can mean that the functionalization occurred since Si is in MPS composition.

#### 3.1.3. X-ray Diffraction (XRD)

Figure 3 shows the X-ray diffractograms of SiO_2_, SiO_2_-Ag, and SiO_2_-Ag-MPS nanoparticles. There was a band around 2θ = 24° regarding SiO_2_ related to the amorphous structure of the material [30]. The presence of five peaks for the SiO_2_-Ag samples corresponding to metallic and cubic silver can be observed, corroborating the incorporation of AgNPs on SiO_2_ surfaces, as shown by Figure 1 and Table 1. The peaks were assigned to the following diffraction planes by JCPDS sheet 01-087-0597: 2θ = 38.11° (111), 2θ = 44.29° (200), 2θ = 64.44° (220), 2θ = 77.39° (311), and 2θ = 81.54° (222).

After the functionalization steps using MPS, there was no significant change in the diffractograms when compared to SiO_2_-Ag. No peaks referring to the formation of silver oxide were identified in the diffractograms, indicating only the presence of metallic silver even after surface modification. According to the results, the functionalization did not generate changes in the crystal structure of the SiO_2_-Ag samples.

#### 3.1.4. Fourier Transform Infrared Spectroscopy (FT-IR)

Figure 4 shows the FT-IR spectra of the SiO_2_, SiO_2_-Ag, and SiO_2_-Ag-MPS samples. There were bands related to Si and O bonds for SiO_2_. The band around 794 cm^−1^ is related with bending mode (torsion) of Si-O-Si bonds [31]. The band at 951 cm^−1^ is related to Si-OH asymmetric stretching [32,33], while the band at 1054 cm^−1^ is associated with Si-O-Si asymmetric stretch vibration [31]. The shoulder around 1169 cm^−1^ refers to the asymmetrical stretch of the Si-O bond [31] and the band at 1627 cm^−1^ is associated with vibrations of adsorbed water molecules [34].

New bands at 1328 and 1383 cm^−1^ were observed after incorporating AgNPs (SiO_2_-Ag). These bands were related to the symmetrical stretching of the N=O bond of the NO_2_ ion which could be from AgNO_3_ used in the coating process [33,35,36]. Bands at 1054 to 1080 cm^−1^ for this sample referring to the asymmetrical stretching of the Si-O-Si were also observed, while Si-OH asymmetric stretching caused a reduction of band intensity at 955 cm^−1^. Such phenomena may be related to the adhesion of silver to the SiO_2_ surface. During the coating process, the Si-O-Si and Si-OH bonds broke to form Si-O-Ag^δ+^ [37]. As a result, the intensity of the band related to Si-O-Si and Si-OH bonds is reduced when compared to pristine SiO_2_ [33].

SiO_2_-Ag bands were still present after functionalization using MPS (SiO_2_-Ag-MPS), which indicates preservation in the material’s primary structure since the modification was only on its surface. However, new bands emerged at 1295, 1321, and 1717 cm^−1^. The bands at 1295 and 1321 cm^−1^ were related to the symmetrical and asymmetrical stretching of the C-O and C-O-C bonds, respectively [38,39]. On the other hand, the band at 1717 cm^−1^ refers to C=C stretching [39,40]. The new bands observed were related to the carbon structure present in the MPS. The band around 930 cm^−1^, referring to Si-OH bonds, had an intensity increase due to Si-OH groups’ presence in the MPS [41,42].

Table 4 presents the FT-IR band assignments for the samples under study.

New SiO_2_-Ag-MPS bands indicated that functionalization with MPS occurred. The surface modification with MPS happened due to a chemical reaction between the hydroxyl groups of the silica surfaces and the Si-OH formed in the groups’ hydrolysis of the MPS (OR’)_3_. After the modification performed with MPS, acrylic groups (Si-(CH_2_)_3_CO_2_(CH_3_)C=CH_2_) were exposed on the surface of the nanoparticles, facilitating its interaction with PMMA.

### 3.2. Minimum Inhibitory Concentration for Candida albicans (SC 5314)

There was no growth inhibition of *C. albicans* for the *SiO*_2_ nanoparticles at any of the concentrations tested, and fungal growth happened in all seeded wells, except in well 12 (control of sterility of the culture medium). The minimum inhibitory concentration observed for SiO_2_-Ag nanoparticles in the presence of *C. albicans* was 1000 μg/mL (Figure 5a). A bacteriostatic effect occurred at the concentration of 2000 μg/mL. For the SiO_2_-Ag-MPS nanoparticles, the MIC effect occurred at the concentration of 2000 μg/mL (Figure 5b). A bacteriostatic effect also happened in the highest tested concentration of 8000 μg/mL.

### 3.3. Microstructural Characterization of Nanocomposites

#### 3.3.1. Analysis of the Surface of the Nanocomposite by Scanning Electron Microscope (SEM)

Figure 6 and Figure 7 show the nanocomposite surface. There was a relevant surface modification after nanoparticle incorporation, mainly in the G2 and R2 groups. There was a homogeneous distribution of nanoparticles on the material’s surface for both the glaze and the reliner. The homogeneous distribution occurred because of the MPS, ensuring superior dispersion.

#### 3.3.2. Contact Angle and Surface Energy Analysis by Goniometer

Table 5 shows the means and standard deviations of contact angle regarding glaze nanocomposites.

There was a statistically significant difference (*p*-value = 0.001) comparing the CG, G1, and G2. There was a superior contact angle for the G2 group, with 2.5 wt% of nanoparticles.

Table 6 shows the means and standard deviations regarding soft reliner nanocomposites.

There was also a statistically significant difference (*p*-value = 0.000) comparing the CR, R1, and R2 groups. There was a superior contact angle for the R2 group, with 2.5 wt% of nanoparticles.

The CG and CR groups had a smaller contact angle when compared to the other experimental groups, resulting in higher wettability and higher surface energy, which facilitates interaction with the medium. The silver-coated silica nanoparticles increased hydrophobicity for both the glaze and the reliner in the experimental groups. Thus, there was a reduction in surface energy, making the surface less permeable.

### 3.4. Microbiological Characterization of Nanocomposites—Evaluation of the Anti-Biofilm Activity of Candida albicans

Table 7 showed the CFU/specimen values for the CG, G1, and G2 (*p*-value not adjusted for ties: 0.245/*p*-value adjusted for ties: 0.244). There was no statistical difference for the presence of nanoparticles, regardless of their concentration, given by the Kruskal–Wallis test.

Table 8 shows the CFU/specimen values for the CR, R1, and R2 groups (*p*-value not adjusted for ties: 0.265/*p*-value adjusted for ties: 0.264). There was no statistical difference for the presence of nanoparticles, regardless of their concentration, given by the Kruskal–Wallis test. CR presented the lowest CFU/specimen value, so the nanoparticles did not generate benefits in terms of fungal reduction for the soft reliner.

Figure 8 show the biofilms formed on a sample from each experimental group. The decline in the CFU/specimen regarding the glaze determined a percentage reduction. The G1 group obtained a percentage reduction of CFU/specimen of 64% compared to the CG. On the other hand, the G2 group had a 12% reduction in CFU/specimen compared to the CG. The R1 and R2 experimental groups obtained higher CFU/specimen values for the reliner compared to the CR group, so there was no fungal reduction.

## 4. Discussion

Acrylic resin is a material susceptible to absorbing small amounts of water when placed in a humid environment, tending to absorb solvents [2,5]. Incorporating SiO_2_-Ag nanoparticles aimed to create an impermeable protective layer to apply to the acrylic resin surface. To do so, the nanoparticles were mixed in glaze and soft reliner, which are polymeric materials with an affinity to bond to the denture’s acrylic resin [2].

Despite its antimicrobial effect, AgNPs can also be cytotoxic to the human body depending on their size, tissue allocation, cellular absorption, surface electric charge, and infiltration competence [43,44,45,46]. Several studies have shown that AgNPs’ cytotoxicity exhibits a dose and time-dependence, being reduced by surface coating techniques [47]. At low concentrations, AgNPs have several applications in different areas of dentistry, such as endodontics, dental prosthesis, implantology, and restorative dentistry. Their incorporation aims to prevent or reduce bacterial and fungal colonization in dental materials, improving patient’s oral health and quality of life [48]. Cytotoxicity was not evaluated in this study, as the nanoparticle had a SiO_2_ core, being described in other studies as a stabilizer of antimicrobial agents, reducing the cytotoxicity of the materials [21,49].

It has been described in other studies that the bactericidal activity of AgNPs is dependent on their size, shape, and quantity [50,51]. Silver nanoparticle size plays a paramount role in bacterial death. Smaller-sized nanoparticles have superior potential for toxicity and slow release of atoms and ions on the surface. The present mechanism of bacterial growth inhibition by AgNPs is not very clear, but their inhibitory action is related to the controlled release of Ag^+^ ions [21]. Thus, the present study aimed to synthesize stable SiO_2_ nanoparticles to act as carrying material to perform the controlled release of Ag^+^. After characterization of the material by FE-SEM, it was possible to observe SiO_2_ nanoparticles that ranged in size from 418 to 502 nm with clustered AgNPs with diameter in the range of 7 to 25 nm. Even though the nanoparticle sizes were adequate in the present study, synthesis methodology improvements must reduce the nanoparticle size. As shown by Devi et al., silica/silver core-shell nanoparticles can act as a antimicrobial agent due to the SiO_2_ high surface area, allowing the loading of large amounts of antimicrobial materials and the slow release of bactericidal agents for an extended period [21].

SEM-EDX characterization showed that the results obtained by this study agreed with other studies already described in the literature [8,24,29,48]. The EDX characterization of SiO_2_, SiO_2_-Ag, and SiO_2_-Ag-MPS nanoparticles showed a majority of Si, O, and Ag chemical elements. Variation in the mass or atomic percentage of each element was related to the characteristic of each sample. As expected, Si and O elements were the major components for the SiO_2_ sample. A large amount of silver was observed after the silver coating, indicating its loading on the SiO_2_ surfaces. The silanization process also made some changes in the EDX results, increasing the number of Si element when compared to the SiO_2_-Ag sample, which is related to the nanoparticle functionalization. The EDX spectra can also be related to the spectra described by Lu [49], which also found characteristic Si, O, and Ag spectra for the silver-decorated mesoporous silica nanoparticles.

There were similar results found in other studies regarding XRD characterization [8,49]. The present study detected an amorphous structure of SiO_2_ by XRD. Gholami and Lu [30,49] found the same amorphous structure of SiO_2_ [30,49]. There was a change in the crystallographic structure in the nanoparticles incorporated with silver, in which it was possible to observe the peaks related to silver [8,49] for both non-functionalized and functionalized material.

FT-IR characterization showed similar results compared to the findings of Menezes, Yu, Sakthisabarimoorthi and Hu [24,31,32,33]. As described in the Results section, the chemical bonds of H_2_O, Si-O, Si-O-Si, and Si-OH represented in the SiO_2_ bands were also found by Yu, Sakthisabarimoorthi and Hu [31,32,33]. New chemical bonds consisting of N=O were assembled for the SiO_2_-Ag nanoparticles, which came from AgNO_3_ used in the silver-coating process [33]. After functionalization by MPS, new bands related to chemical bonds between C-O, C-O-C [38,39], and C=C [39,40] appeared. These news carbon bands came from MPS, which means that functionalization occurred. In addition, there was an increase in Si-OH chemical bonds due to the presence of Si-OH groups in the MPS [41,42]. Menezes et al. found the same new bands after functionalization with MPS [24], in which they identified acrylic and silanol groups as being responsible for the best chemical interaction with the acrylic resin.

The MPS organosilane was used as coupling agent due to its ability to couple with organic matrix, such as acrylic-based polymers, and to inorganic fillers, like the SiO_2_-Ag hybrid nanoparticles [52]. The inserted functional groups, especially the methacryloxy group, are expected to improve both the dispersion and interaction of the nanofillers in the acrylic matrix. According to the manufacturer, the soft reliner is constituted of ethyl methacrylate, while the glaze is constituted of methyl methacrylate. Thus, the methacrylate group is present in soft reliner, glaze, and MPS structure and this molecular structure similarity favors the interaction between the components [53]. In addition, the C=C bond in the MPS methacrylate group can copolymerize with the C=C bond of ethyl methacrylate (soft reliner) and methyl methacrylate (glaze) during the polymerization reactions of the dental materials [52].

The toxicity of MPS monomer in living tissues remains uncertain due to the lack of studies in the area [54,55]. However, in the present work, MPS is used as functionalization agent for SiO_2_-Ag nanoparticles, being used in a very low concentration. In addition, as cited, MPS methacrylate groups are copolymerized with the soft reliner and glaze monomers, reducing its toxicity.

After the microstructural characterization, SiO_2_, SiO_2_-Ag, and SiO_2_-Ag-MPS were submitted to MIC assay for *C. albicans*. According to the results, the pure SiO_2_ nanoparticle had no antifungal action, observing growth in all wells regardless of the concentration reduction. Lu et al. found that SiO_2_ has no antifungal effect, but the microorganisms tested were *S. aureus* and *E. coli* [49]. Qasim et al. found that AgNPs embedded with mesoporous SiO_2_ nanospheres had a promising antifungal effect against *C. albicans*, and it is dose dependent as increasing concentrations decreases fungal growth [56]. The SiO_2_-Ag and SiO_2_-Ag-MPS nanoparticles obtained MIC values for *C. albicans* of 1000 μg/mL and 2000 μg/mL, respectively. Liao et al. [57] reported MIC values ranging from 1406–5625 μg/mL, but the microorganism studied was *P. aeruginosa*. In another study, the antifungal effect of AgNPs was limited against *T. harzianum*, and there was no antifungal effect for *G. candidum*, even at the highest AgNP concentration. However, these authors found a MIC of 328.05 μg/mL against *M. phaseolina*, *A. alternata*, and *F. oxysporum* [58], differing from the present study in which there was an antifungal effect for *C. albicans*.

SiO_2_-Ag-MPS nanoparticles were incorporated into acrylic resin to improve the dispersion of nanoparticles over resin bulk, as described by Menezes et al. [24]. The nanocomposites’ characterization was performed by FE-SEM and by goniometry. FE-SEM indicates that the surface modification occurred due to the nanoparticle incorporation into the acrylic matrix for both glaze and soft reliner, regardless of concentration (1.0 wt% and 2.5 wt%). SEM images of the G2 and R2 group specimens showed a superior number of nanoparticles on the surface compared to the CG, G1 and CR, and R1 groups.

SiO_2_-Ag-MPS nanoparticles significantly reduce the materials’ surface energy for the contact angle. Therefore, the materials’ permeability also decreased compared to the control groups (CG and CR). These results differed from the results reported in the literature, whose incorporation of SiO_2_-Ag nanoparticles did not change the material’s contact angle [59]. The difference among these studies has its probable cause related to the MPS utilization, since the present study employed MPS. The results of the present study agree with the findings of Ziabka et al. [60], in which the contact angle had a statistically significant increase after incorporating the *Ag* nanoparticles.

As stated, the biofilm results were only promising for the glaze. Although the reduction was not statistically significant, there was a 64% reduction in the CFU/specimen count between CG and G1. In another study, a reduction of about 50% in the biofilm formation was observed for the nanocomposites with incorporation of silver nanowires [61]. Li et al. [62] found that there was a biomass percentage reduction of *C. albicans* with the use of a AgNPs suspension on a denture base made of acrylic resin. In this study, the higher the concentration of the antifungal agent, the greater the biomass percentage reduction [62]. These findings did not agree with the present study in which antifungal agent increase did not reduce the CFU/specimen growth, and G1 had a superior CFU/specimen. It is noteworthy that there are differences between the materials used in the studies for both the antifungal agent and for the polymeric material. Jasiorski et al. [27] found that 2.5 wt% of SiO_2_-Ag nanoparticles in a textile fiber reduced microorganism growth for *S. aureus* and *E. coli*. No growth reduction happened for the groups with 1.0 wt% of the nanoparticles for both microorganisms [27]. This result differs from the present study since G1 had the lowest CFU/specimen count, although the nanoparticles were tested in different materials, constituting a possible cause of the difference in results.

Since soft reliner is a porous material and there is biofilm incorporation over its surface in its clinical application, it is necessary to incorporate other materials in its bulk. Silver nanoparticles were selected because of the previously reported antifungal effect [63,64]. SiO_2_ was the vehicle that carried the silver nanoparticles, aiming to reduce its toxicity and to regulate its release, which could extend the antimicrobial effect [21,49].

The results found in this study promote further understanding of SiO_2_-Ag silanization mechanism using MPS organosilane and its effects on antifungal activity of acrylic-based dental matrices. The results suggested that the sinalization using MPS can produce nanocomposites with superior antifungal activity against *C. albicans*. The release of silver from the acrylic matrix, and consequently antifungal action, can be improved by developing new studies regarding the controlled release of AgNPs by SiO_2_ surfaces. Moreover, interference of the functionalization process in the SiO_2_-Ag hybrid nanofiller must be studied more. Although the functionalization improved the interaction and dispersion of the nanofillers with the polymeric matrix, the effect of this functionalization on the release of AgNPs must be investigated. In addition, it is crucial to carry out further studies to improve the color of the material, and to evaluate the cytotoxicity and mechanical properties of the produced materials.

## 5. Conclusions

Within the study limitations, it can be concluded that the synthesis and characterization of SiO_2_, SiO_2_-Ag, and SiO_2_-Ag-MPS nanoparticles obtained morphological characteristics at the nanoscale. The crystal structure in nanoparticles remained stable even after functionalizing the material with the MPS group. There was a reduction in fungal growth in the wells that received the SiO_2_-Ag and SiO_2_-Ag-MPS nanoparticles at the concentration of 1000 and 2000 μg/mL, respectively. There was a significant change in the surface energy and wettability for the nanocomposites, glaze, and soft reliner. The G2 and R2 groups had the most significant increase. The G1 group had a 64% reduction of CFU/specimen compared to the CG group, but still not statistically significant. There was a 12% reduction for the G2 group when compared to the CG. The development of a glaze for resinous materials may have promising antifungal potential, as it demonstrated a 64% reduction in CFU/specimen. In addition, there was a significant reduction in the wettability when the nanoparticle was present.

## Figures and Tables

**Figure 1 polymers-14-04306-f001:**
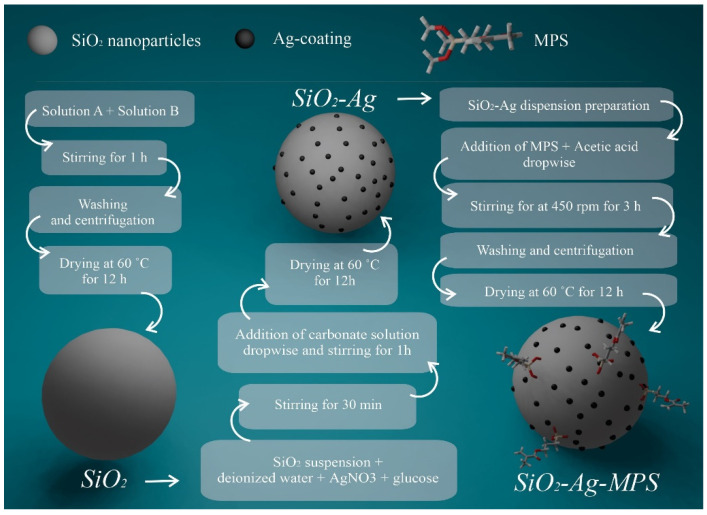
Schematic diagram of the synthesis of SiO_2_, SiO_2_-Ag and SiO_2_-Ag-MPS.

**Figure 2 polymers-14-04306-f002:**
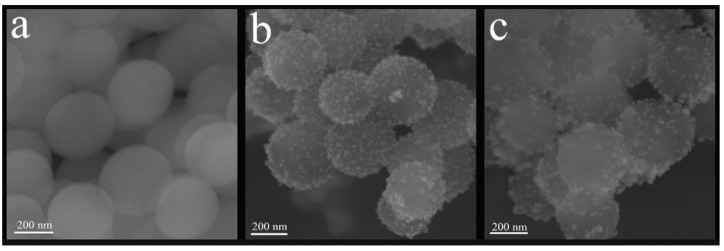
Images obtained by FE-SEM of the samples (**a**) pure SiO_2_, (**b**) non-functionalized SiO_2_-Ag, (**c**) SiO_2_-Ag functionalized by MPS.

**Figure 3 polymers-14-04306-f003:**
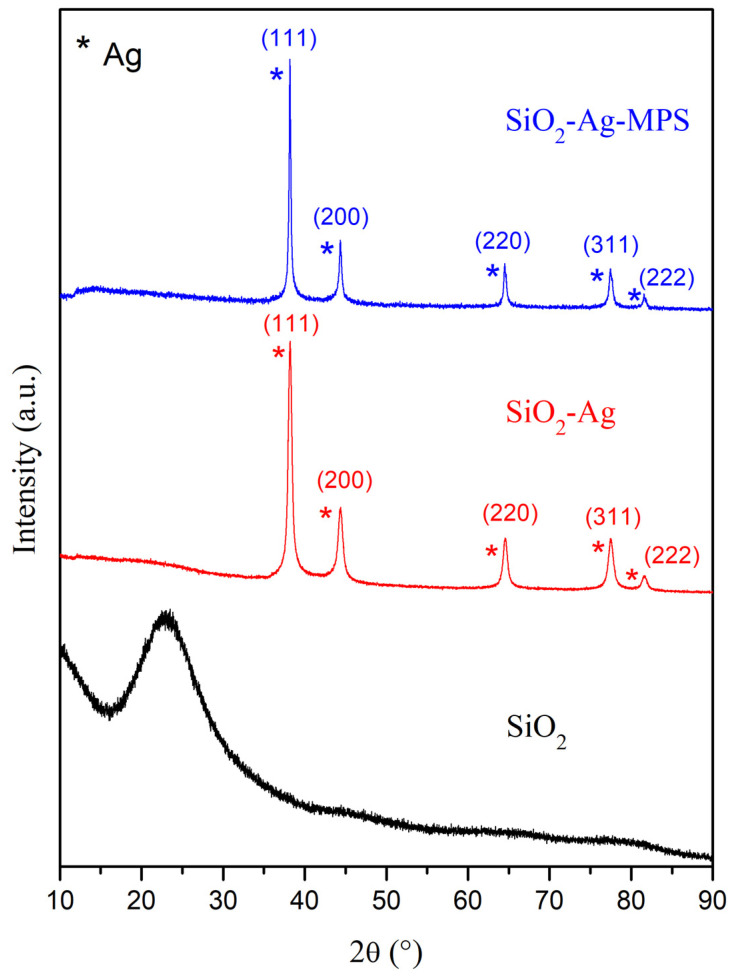
XRD analysis of SiO_2_, SiO_2_-Ag, and SiO_2_-Ag-MPS nanoparticles. The peaks highlighted by the asterisk (*) refer to the metallic silver (Ag).

**Figure 4 polymers-14-04306-f004:**
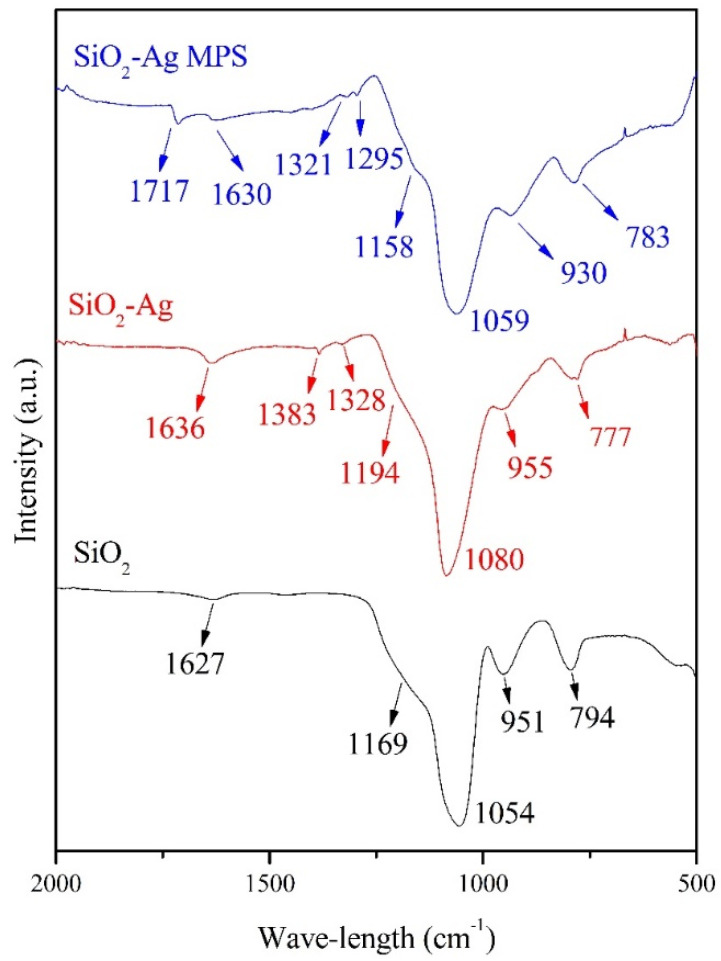
FT-IR spectra of SiO_2_, SiO_2_-Ag, and SiO_2_-Ag-MPS nanoparticle samples from 2000 to 500 cm^−1^.

**Figure 5 polymers-14-04306-f005:**
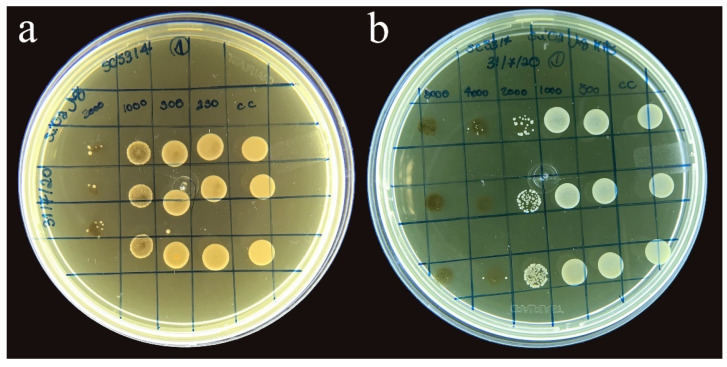
(**a**) Minimum inhibitory concentration of SiO_2_-Ag nanoparticles; (**b**) Minimum inhibitory concentration of SiO_2_-Ag-MPS nanoparticle.

**Figure 6 polymers-14-04306-f006:**
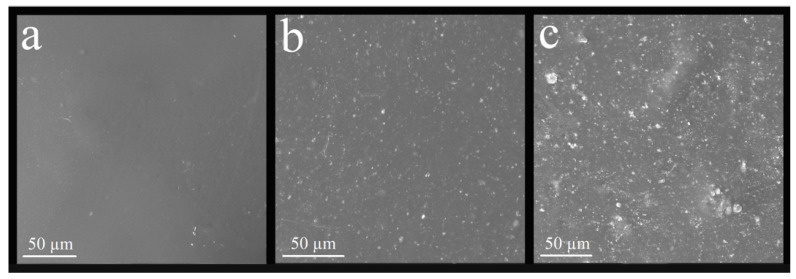
Images obtained by SEM-EDX of the surface of the nanocomposites of the experimental groups relative to the glaze: (**a**) group CG, (**b**) group G1, and (**c**) group G2.

**Figure 7 polymers-14-04306-f007:**
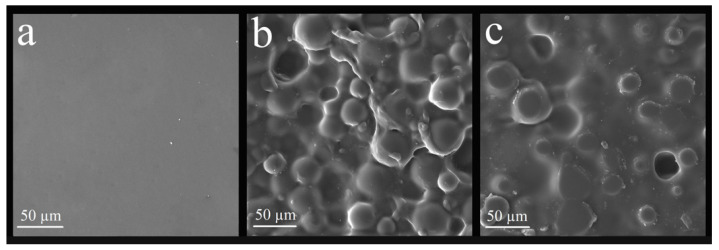
Images obtained by SEM-EDX of the surface of the nanocomposites of the experimental groups relative to the reliner: (**a**) CR group, (**b**) R1 group, and (**c**) R2 group.

**Figure 8 polymers-14-04306-f008:**
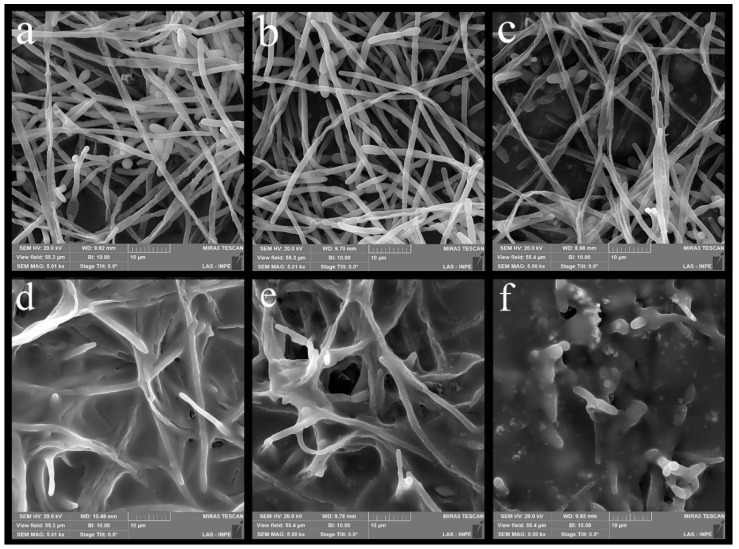
Images obtained by MEV-FEG of the biofilm formed on the surface of the nanocomposites: (**a**) CG group, (**b**) G1 group, (**c**) G2 group; (**d**) CR group, (**e**) R1 group, (**f**) R2 group.

**Table 1 polymers-14-04306-t001:** Division of experimental groups according to the concentration of SiO_2_-Ag-MPS nanoparticles for the glaze.

Groups (*n* = 12)	Description
CG	Glaze + Soft reliner
G1	(Glaze + SiO_2_-Ag-MPS nanoparticles 1 wt%) + Soft reliner
G2	(Glaze + SiO_2_-Ag-MPS nanoparticles 2.5 wt%) + Soft reliner

**Table 2 polymers-14-04306-t002:** Division of experimental groups according to the concentration of SiO_2_-Ag-MPS nanoparticles for the reliner.

Groups (*n* = 12)	Description
CR	Soft reliner
R1	Soft reliner + SiO_2_-Ag-MPS 1 wt% nanoparticles
R2	Soft reliner + SiO_2_-Ag-MPS 2.5 wt% nanoparticles

**Table 3 polymers-14-04306-t003:** Means of mass percent (wt%) and atomic percent (at%) values of SiO_2_, SiO_2_-Ag, and SiO_2_-Ag-MPS nanoparticles by SEM-EDX analysis.

Element	SiO_2_		SiO_2_-Ag		SiO_2_-Ag-MPS	
	wt%	at%	wt%	at%	wt%	at%
C	1.2	1.9	6.7	13.5	2.0	4.3
O	54.1	66.7	4.5	62.9	35.2	61.5
Si	44.7	31.5	18.6	16.1	23.6	23.9
Ag	-	-	33.2	7.5	38.2	10.4

**Table 4 polymers-14-04306-t004:** Assignment of FT-IR bands for SiO_2_, SiO_2_-Ag, and SiO_2_-Ag-MPS samples.

Sample	Bands (cm^−1^)	Assignment	Reference
	1627	H_2_O vibration	[34]
SiO_2_	1169	Asymmetric stretching vibration of the Si-O bond	[31]
	1054	Asymmetric stretching vibration of Si-O-Si bonds	[31]
	951	Si-OH asymmetric stretching vibration	[32,33]
	794	Bending mode (torsion)	[31]
SiO_2_-Ag	1636	H_2_O vibration	[34]
	1383	Symmetric stretching vibration of N=O bond	[33]
	1328	Symmetric stretching vibration of N=O bond	[33]
	1194	Asymmetric stretching vibration of the Si-O bond	[31]
	1080	Asymmetric stretching vibration of Si-O-Si bonds	[31]
	955	Si-OH asymmetric stretching vibration	[32,33]
	777	Bending mode (torsion)	[31]
SiO_2_-Ag-MPS	1717	Stretching vibration of C=C	[39,40]
	1630	H_2_O vibration	[34]
	1321	Symmetric stretching vibration of C-O-C bond	[38,39]
	1295	Symmetric stretching vibration of C-O bond	[38,39]
	1158	Asymmetric stretching vibration of the Si-O bond	[31]
	1059	Asymmetric stretching vibration of Si-O-Si bonds	[31]
	930	Si-OH asymmetric stretching vibration	[31]
	783	Bending mode (torsion)	[31]

**Table 5 polymers-14-04306-t005:** Mean and standard deviation for the contact angle with Tukey’s test (α = 0.05) for the glaze.

Groups	Mean ± Standard Deviation	Tukey’s Test
CG	53.3 ± 11.0	B
G1	57.2 ± 10.6	B
G2	73.7 ± 12.1	A

**Table 6 polymers-14-04306-t006:** Mean and standard deviation for the contact angle with Tukey’s test (α = 0.05) for the reliner.

Groups	Mean ± Standard Deviation	Tukey’s Test
CR	35.6 ± 10.9	B
R1	44.8 ± 7.8	B
R2	57.9 ± 10.7	A

**Table 7 polymers-14-04306-t007:** CFU/specimen values for the glaze.

Groups	*n*	Median	Ave Rank	Z-Value
CG	12	1,300,000	20.7	0.87
G1	12	470,000	14.3	−1.68
G2	12	1,150,000	20.5	0.81
Overall	36		18.5	

**Table 8 polymers-14-04306-t008:** CFU/specimen values for the reliner.

Groups	*n*	Median	Ave Rank	Z-Value
CR	12	470,000	17.0	−0.59
R1	12	475,000	16.0	−1.02
R2	12	1,150,000	22.5	1.61
Overall	36		18.5	

## Data Availability

Not applicable.

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
