# Peer review of "Silver-Coated Silica Nanoparticles Modified with MPS: Potential Antimicrobial Biomaterials Applied in Glaze and Soft Reliner"

_polymers, 2022, doi:10.3390/polym14204306_

Round 1

Reviewer 1 Report

Dear Editor,

I have read the manuscript entitled: “Silver-coated silica nanoparticles modified with MPS: potential antimicrobial biomaterials applied in glaze and soft reline” and I would like to address following suggestions to the authors:

1-Please add some lines to indicate the novelty of your study, compare the results with that of the literature and emphasize the novelty of this study.

2-Line 82: please create new tables and put data in it (for example table 1, 2, 3) and replace Fig.1, Fig.3 and Fig. 7

3-Line 87: please explain about the method.

4-Line 33: please more explain about synthesis nanoparticles.

5-The introduction can be improved by providing a more critical discussion of recent related literature. Discuss the shortcomings of previous work and the gaps and how this work intends to fill those gaps. For example, some papers (Polymer Testing, 93, 106922, (2021); Polymers, 12, 861, (2020); Journal of Sol-Gel Science and Technology, 78, 91-98, (2016); Materials, 15, 5536, (2022)) should be cited.

6-For each research method, it is necessary to expand the discussion and describe by chemical reactions. Please add schematic diagram for this study.

7-Fig.2 (2) please put miller indices in XRD pattern.

8-Please improve quality of figures and arrange them.

Author Response

Reviewer #1:

I have read the manuscript entitled: “Silver-coated silica nanoparticles modified with MPS: potential antimicrobial biomaterials applied in glaze and soft reline” and I would like to address following suggestions to the authors:

1 - Please add some lines to indicate the novelty of your study, compare the results with that of the literature and emphasize the novelty of this study.

Thank you for the observation. As suggested, the highlighted paragraph was added to the introduction:

“This work reports a pioneer silver-coated SiO2 nanoparticle surface modification through silanization reaction using MPS and their incorporation into dental acrylic matrix. Glaze and soft reliner were used as the organic matrix due to their precise indication to reduce inflammation and distribute a balanced load on the oral tissues during dental repairs. A complete characterization of the MPS-silanized silver-coated SiO2 nanoparticles was performed. Then, an unmodified and modified nanofiller were incorporated into glaze and soft reliner polymers. The effect of the nanofillers in the matrix and antifungal activity was evaluated.”

Also, the following highlighted text was added on the discussion:

“As stated, the biofilm results were only promising for the glaze. Although the reduction was not statistically significant, there was a 64% reduction in the CFU/specimen count between CG and G1. In another study, a reduction of about 50% in the biofilm formation was observed for the nanocomposites with incorporation of silver nanowires [60].  Li et al. [61] found that there was a biomass percentage reduction of C. albicans with the use of a AgNPs suspension on a denture base made of acrylic resin. In this study, the higher the concentration of the antifungal agent, the greater the biomass percentage reduction [61]. These findings did not agree with the present study in which antifungal agent increase did not reduce the CFU/specimen growth, and G1 had a superior CFU/specimen. It is noteworthy that there are differences between the materials used in the studies for both the antifungal agent and for the polymeric material. Jasiorski et al. [26] found that 2.5 wt% of SiO2-Ag nanoparticles in a textile fiber reduced microorganism growth for S. aureus and E. Coli. No growth reduction happened for the groups with 1.0 wt% of the nanoparticles for both microorganisms [26]. This result differs from the present study since G1 had the lowest CFU/specimen count, although the nanoparticles were tested in different materials, constituting a possible cause of the difference in results.

Since soft reliner is a porous material and there is biofilm incorporation over its surface in its clinical application, it is necessary to incorporate other materials in its bulk. Silver nanoparticles were selected because of the previously reported antifungal effect [62,63]. SiO2 was the vehicle that carried the silver nanoparticles, aiming to reduce its toxicity and to regulate its release, which could extend the antimicrobial effect [21,48].”

2 - Line 82: please create new tables and put data in it (for example table 1, 2, 3) and replace Fig.1, Fig.3 and Fig. 7

Thank you for the observation. As suggested, we replaced the figures 1, 3, and 7 for the tables 1 to 8. Also, the table of “materials used” was removed, and its information was included in the text.

3 - Line 87: please explain about the method.

Thank you for the suggestion. The following highlighted text was added in “Synthesis of SiO2 nanoparticles”:

“The SiO2 nanoparticles were produced using the Stöber method [26], which consists of producing monodisperse suspension of silica spheres in the colloidal size range through the hydrolysis and controlled condensation of tetraethylorthosilicate (TEOS, 98.0 %, Si(OCH3)4, Sigma) in an alcoholic medium [25].”

4 - Line 33: please more explain about synthesis nanoparticles.

Thank you for the suggestion. The following highlighted text was included in “Synthesis of SiO2 nanoparticles” to clarify the synthesis method:

“For this purpose, two different solutions were prepared separately: (1) “Solution A”: 9 mL of TEOS and 55 mL of ethyl alcohol (98.8 %, C2H6O, Synth); and (2) “Solution B”: 30 mL of deionized water, 55 mL of ethyl alcohol, and 5 mL of ammonium hydroxide (24.0 %, NH4OH, Synth). “Solution A” and “Solution B” were separately kept in magnetic stirring until homogenization. Then, “Solution B” was totally poured into “Solution A” and the mixture was kept under constant stirring for one hour until the condensation reaction is complete. After this process, the precipitate obtained was washed and centrifuged. Then, the material was dried in an oven at 60 ˚C for 12 h. The sample was labeled as SiO2.”

5 - The introduction can be improved by providing a more critical discussion of recent related literature. Discuss the shortcomings of previous work and the gaps and how this work intends to fill those gaps. For example, some papers (Polymer Testing, 93, 106922, (2021); Polymers, 12, 861, (2020); Journal of Sol-Gel Science and Technology, 78, 91-98, (2016); Materials, 15, 5536, (2022)) should be cited.

Thank you for the suggestion. We added more relevant references in the introduction section to highlight the importance and novelty of this work, including the suggested papers. The following highlighted text and references were included in the “Introduction”:

There is no material available on the dental market which can inhibit microorganisms aggregating on the surface of acrylic resin. Thus, there is a need to create a new material with antimicrobial and antifungal properties whose objective is to form a protective layer capable of inhibiting bacterial and fungal proliferation on the surface of prostheses. In this context, several nanomaterials have been studied by combining biocompatible polymers and inorganic materials, such as SiO2, TiO2, ZnO, and silver, among others [9–13].

Silver nanoparticles (AgNPs) stand out as one of the most intensively investigated systems. They have unique properties, such as excellent electrical conductivity, catalytic activity, non-linear optical behavior, high stability and resistance to oxidation process, and antimicrobial effects, making them potential candidates for applications in different areas including sensors, textiles, electronics, catalysis, food packaging, and medical therapy [14–16]. The use of AgNPs expanded in the dental field because of their bactericidal and bacteriostatic activity against microorganisms such as fungi, bacteria, and viruses [17,18]. Chladek et al. [13] modified silicon soft line polymer with AgNPs (10 to 200 ppm of AgNPs) and evaluated the antifungal efficacy (AFE) of the developed nanocomposites. The results showed AFE varying between 16.4% (10 ppm) to 52.2 % (200 pm) for the examined samples. According to the authors, increasing the amount of AgNPs increased the antifungal behavior.

However, the work of Habibzadeh et al. [19] showed that the addition of AgNPs to silicone soft liner reduced its tensile bond strength. The authors also reported that the release of AgNPs into the oral cavity may increase its toxicity, needing a controlled release to maintain the optimal balance between the biocompatibility and antifungal activity of the nanocomposite. The cited works showed the potential of AgNPs to be used as antibacterial and antifungal materials; however, further studies should be carried out in order to control AgNP toxicity and its effects on polymeric mechanical properties.

In this context, the use of SiO2 nanoparticles have been showing great potential to be used as carrying material for antimicrobial particles. Metal oxides are commonly used due to their stability, biocompatibility, and for being essential minerals for human health [9]. Gad et al. [10] incorporated SiO2 nanoparticles into a soft relining denture material to reduce Candida albicans adhesion. According to the results, the addition of 0.25 and 0.5 % of nano-SiO2 significantly reduced adhesion of C. albicans fungus into acrylic soft liner. However, incorporating 1.0 % of nano-SiO2 resulted in higher C. albicans counts when compared to the 0.25 and 0.5 % samples. The authors suggested that higher nano-SiO2 concentration can lead to their agglomeration in the polymeric matrix.

The combination of antifungal and antibacterial properties of AgNPs with the biocompatibility and stability of SiO2 nanoparticles has great potential. Loading AgNPs in SiO2 surfaces has showed great potential to regulate the solubility and toxicity of Ag+ free ions by controlling their release [20]. SiO2 nanoparticles enable loading large amounts of antimicrobial materials and the slow release of bactericidal agents for a prolonged period, being potentially used as carriers and a core for antimicrobial agents [21]. Many works have been developing techniques to use silver as nanomaterials for biomedical applications by improving its dispersibility, stability, and prolonging the release time. El-Nour et al. studied a methodology to control the release of Ag+ ions by incorporating SiO2 as a nucleus which acts as a silver support [15]. SiO2 is a paramount material for immobilizing nanoparticles on its surface due to its high chemical and thermal stability, chemical inertness, large surface area, and good compatibility with other materials [22,23].  Le et al. [20] evaluated incorporating SiO2-Ag hybrid nanofillers at concentrations of 0.5 to 4.0% wt on water-based acrylic coating. The authors studied the effect of hybrid nanofillers on the abrasion resistance, thermal stability, and antibacterial activity against Escherichia coli. According to the results, incorporating 2.0 % of SiO2-Ag nanofillers simultaneously improves the coating’s mechanical properties, thermal resistance, and antibacterial activity. However, a higher amount of the nanofiller in the matrix (4.0 % wt) showed particle agglomeration, as shown by scanning electron microscopy images, reducing the adhesion and abrasion resistance.

As shown, SiO2 nanoparticles have great potential to be used as carriers and cores for antimicrobial agents, such as AgNPs. However, many works have indicated the tendency for agglomeration and reduction of compatibility of these nanomaterials with organic matrices, especially when the amount of nanofiller is superior to 2.0 wt%. The modification of the SiO2-AgNPs nanofiller surfaces is an alternative to improve their interaction and dispersion in the polymeric matrix. γ–methacryloxypropyltrimethoxysilane (MPS) is an important silane which is widely used as a functionalization agent to promote interfacial interactions between metal oxide nanoparticles and a polymeric matrix, such as dental acrylic resins. The silanization mechanism occurs by M− O− Si (M = metal) bonds between MeO-NPs and silane coupling agents [24].

Several works have shown improvements in silanization of nanofillers used in acrylic-based matrices for dental applications.  Menezes et al. [24] modified the surface of silver vanadate nanorods (AgVO3) using MPS to enhance the dispersion and interaction of this filler with polymethyl methacrylate (PMMA). The authors showed that nanocomposites produced with MPS-modified AgVO3 had higher Shore-D hardness and impact strength values than a nanocomposite with pristine AgVO3 due to the improved dispersion and interaction.

This work reports a pioneer silver-coated SiO2 nanoparticle surface modification through silanization reaction using MPS and their incorporation into dental acrylic matrix. Glaze and soft reliner were used as the organic matrix due to their precise indication to reduce inflammation and distribute a balanced load on the oral tissues during dental repairs. A complete characterization of the MPS-silanized silver-coated SiO2 nanoparticles was performed. Then, an unmodified and modified nanofiller were incorporated into glaze and soft reliner polymers. The effect of the nanofillers in the matrix and antifungal activity was evaluated. Therefore, the aim of this study was to synthesize SiO2-Ag MPS nanoparticles, to characterize them, to test the minimum inhibitory concentration against Candida albicans, and to perform the anti-biofilm test against Candida albicans.”

6 - For each research method, it is necessary to expand the discussion and describe by chemical reactions. Please add schematic diagram for this study.

Thank you for the suggestion. Regarding the discussion describing chemical reactions, we added the following paragraph to explain these reactions and their role in the material’s systhesis:

“The synthesis of SiO2 was performed using the Stöber method. The simplified reactions of hydrolysis (1) and condensation (2) are [26]:    

(1)

(2)

In this reaction, TEOS is controlled and hydrolyzed (1) in a medium compounded by ethanol, producing Si(OH)4. Then the dispersed silanol phase is condensed in a polymerization reaction (2) when the SiO2 nanoparticles are obtained.

A previous work regarding the Ag coating reaction on SiO2 nanoparticles reports that the addition of silver nitrate, glucose, and Na2CO3 to SiO2 suspension enables forming [Ag(NH3)2]+ ions which are electrostatically attracted to silanol groups. Then the ionic Ag is reduced to metallic Ag on SiO2 surfaces, obtaining the SiO2-Ag structure [18].

The reaction of MPS (RSi-OCH3) bonding to SiO2 was also already discussed in the literature. The hydroxyl groups on the SiO2 surface react with the OCH3 groups available in MPS molecules by acid hydrolysis [18,27]. Then, the obtained products are RSi-O-Si and methanol (3) - which are withdrawn in the washing process and not present in the final product (SiO2-Ag-MPS).

               (3)

Also, it can be seen below the schematic diagram of the synthesis.

The full schematic diagram of the synthesis is displayed in Figure 1.

Figure 1: Schematic diagram of the synthesis of SiO2, SiO2-Ag and SiO2-Ag-MPS.

7 - Fig.2 (2) please put miller indices in XRD pattern.

Thank you for the observation. As suggested, we included the miller indices in the XRD pattern:

8 - Please improve quality of figures and arrange them.

Thank you for the observation. All Figures and Tables were properly and individually presented.

Reviewer 2 Report

The paper entitled "SILVER-COATED SILICA NANOPARTICLES MODIFIED WITH MPS: POTENTIAL ANTIMICROBIAL BIOMATERIALS APPLIED IN GLAZE AND SOFT RELINE" is an attempt to carried out a study regarding the the influence of silver nanoparticles and functionalization of silica on the antimicrobial activity for applications in glaze and soft relining materials.

Overall, the content of the paper is poor. It is not clear the aim and the results do not support their suitability for such applications. The novelty of the paper is also missing. Why are necessary such materials? The authors must identify the problem of the already existing materials and then look for the solutions.

 It seems rather a study of nanomaterials and nanocomposites than of polymers, so that the inclusion of such a paper in a Polymers journal is completely inappropriate. The authors also must develop some new polymers to complete these studies. The motivation to choose SiO2 nanoparticles is unclear. Why the authors choose to functionalize these particles with MPS is also unclear.

The Materials and methods section is inappropriately presented as figure 1.

The authors also claim that they obtained SiO2 coated with Ag NPs. They must prove that.- lines 98-105

The section 2.6 is ambiguous presented.

The section 3.1.4. must be improved. There are some observations made by assumptions at lines 281-283, 286-288, 292-294. Also the information presented in Figure 3 are a combination between SEM, XRD and FTIR data and must be changed.

Figure 4 does not support the comments, must be replaced or improved.

At figure 5, a cross-section form each material must be added, not only at the surface.

Section 3.3.2 - add the contact angle values for al samples and then discuss them.

The section 3.4. is unclear presented, some information is missing.

In section 4 there are also some observations made by assumptions at lines 369-370, 382-384, 396-397, 402-403, 406-408, 410-414, 424-426, 475-483 and also in Conclusion section- it is not clear the aim of this paper and the results do not support them.

Based on these consideration my recommendation is Reject.

Author Response

Reviewer #2:

The paper entitled "SILVER-COATED SILICA NANOPARTICLES MODIFIED WITH MPS: POTENTIAL ANTIMICROBIAL BIOMATERIALS APPLIED IN GLAZE AND SOFT RELINE" is an attempt to carried out a study regarding the the influence of silver nanoparticles and functionalization of silica on the antimicrobial activity for applications in glaze and soft relining materials.

1 - Overall, the content of the paper is poor. It is not clear the aim and the results do not support their suitability for such applications. The novelty of the paper is also missing. Why are necessary such materials? The authors must identify the problem of the already existing materials and then look for the solutions.

Thank you for the observation. As suggested, the highlighted paragraph was added to the introduction:

“This work reports a pioneer silver-coated SiO2 nanoparticle surface modification through silanization reaction using MPS and their incorporation into dental acrylic matrix. Glaze and soft reliner were used as the organic matrix due to their precise indication to reduce inflammation and distribute a balanced load on the oral tissues during dental repairs. A complete characterization of the MPS-silanized silver-coated SiO2 nanoparticles was performed. Then, an unmodified and modified nanofiller were incorporated into glaze and soft reliner polymers. The effect of the nanofillers in the matrix and antifungal activity was evaluated.”

Also, the following highlighted text was added on the discussion:

“As stated, the biofilm results were only promising for the glaze. Although the reduction was not statistically significant, there was a 64% reduction in the CFU/specimen count between CG and G1. In another study, a reduction of about 50% in the biofilm formation was observed for the nanocomposites with incorporation of silver nanowires [60].  Li et al. [61] found that there was a biomass percentage reduction of C. albicans with the use of a AgNPs suspension on a denture base made of acrylic resin. In this study, the higher the concentration of the antifungal agent, the greater the biomass percentage reduction [61]. These findings did not agree with the present study in which antifungal agent increase did not reduce the CFU/specimen growth, and G1 had a superior CFU/specimen. It is noteworthy that there are differences between the materials used in the studies for both the antifungal agent and for the polymeric material. Jasiorski et al. [26] found that 2.5 wt% of SiO2-Ag nanoparticles in a textile fiber reduced microorganism growth for S. aureus and E. Coli. No growth reduction happened for the groups with 1.0 wt% of the nanoparticles for both microorganisms [26]. This result differs from the present study since G1 had the lowest CFU/specimen count, although the nanoparticles were tested in different materials, constituting a possible cause of the difference in results.

Since soft reliner is a porous material and there is biofilm incorporation over its surface in its clinical application, it is necessary to incorporate other materials in its bulk. Silver nanoparticles were selected because of the previously reported antifungal effect [62,63]. SiO2 was the vehicle that carried the silver nanoparticles, aiming to reduce its toxicity and to regulate its release, which could extend the antimicrobial effect [21,48].”

2 - It seems rather a study of nanomaterials and nanocomposites than of polymers, so that the inclusion of such a paper in a Polymers journal is completely inappropriate. The authors also must develop some new polymers to complete these studies. The motivation to choose SiO2 nanoparticles is unclear. Why the authors choose to functionalize these particles with MPS is also unclear.

Thank you for the comments. Indeed, the major objective of this work was to develop a new nanocomposite. However, this nanocomposite is based in a polymeric matrix and devoted to biomedical applications, being one of the scopes of the “Polymers” journal, as cited on the website (https://www.mdpi.com/journal/polymers/about):

Polymer Composites and Nanocomposites: Design of polymer composites and nanocomposites, fibrer-reinforced polymers, biomedical composites, structural composites, multifunctional composites, biomimetic and eco composites, polymer foams, smart composites, modelling of polymer composites and nanocomposites, self-healing of polymer composites and nanocomposites, life cycle assessment of polymer composites and nanocomposites.”

Regarding the use of SiO2 nanoparticles and the MPS functionalization process, the highlighted sentences were included in the introduction to clarify the work motivation:

“Silver nanoparticles (AgNPs) stand out as one of the most intensively investigated systems. They have unique properties, such as excellent electrical conductivity, catalytic activity, non-linear optical behavior, high stability and resistance to oxidation process, and antimicrobial effects, making them potential candidates for applications in different areas including sensors, textiles, electronics, catalysis, food packaging, and medical therapy [14–16]. The use of AgNPs expanded in the dental field because of their bactericidal and bacteriostatic activity against microorganisms such as fungi, bacteria, and viruses [17,18]. Chladek et al. [13] modified silicon soft line polymer with AgNPs (10 to 200 ppm of AgNPs) and evaluated the antifungal efficacy (AFE) of the developed nanocomposites. The results showed AFE varying between 16.4% (10 ppm) to 52.2 % (200 pm) for the examined samples. According to the authors, increasing the amount of AgNPs increased the antifungal behavior.

However, the work of Habibzadeh et al. [19] showed that the addition of AgNPs to silicone soft liner reduced its tensile bond strength. The authors also reported that the release of AgNPs into the oral cavity may increase its toxicity, needing a controlled release to maintain the optimal balance between the biocompatibility and antifungal activity of the nanocomposite. The cited works showed the potential of AgNPs to be used as antibacterial and antifungal materials; however, further studies should be carried out in order to control AgNP toxicity and its effects on polymeric mechanical properties.

In this context, the use of SiO2 nanoparticles have been showing great potential to be used as carrying material for antimicrobial particles. Metal oxides are commonly used due to their stability, biocompatibility, and for being essential minerals for human health [9]. Gad et al. [10] incorporated SiO2 nanoparticles into a soft relining denture material to reduce Candida albicans adhesion. According to the results, the addition of 0.25 and 0.5 % of nano-SiO2 significantly reduced adhesion of C. albicans fungus into acrylic soft liner. However, incorporating 1.0 % of nano-SiO2 resulted in higher C. albicans counts when compared to the 0.25 and 0.5 % samples. The authors suggested that higher nano-SiO2 concentration can lead to their agglomeration in the polymeric matrix.

The combination of antifungal and antibacterial properties of AgNPs with the biocompatibility and stability of SiO2 nanoparticles has great potential. Loading AgNPs in SiO2 surfaces has showed great potential to regulate the solubility and toxicity of Ag+ free ions by controlling their release [20]. SiO2 nanoparticles enable loading large amounts of antimicrobial materials and the slow release of bactericidal agents for a prolonged period, being potentially used as carriers and a core for antimicrobial agents [21]. Many works have been developing techniques to use silver as nanomaterials for biomedical applications by improving its dispersibility, stability, and prolonging the release time. El-Nour et al. studied a methodology to control the release of Ag+ ions by incorporating SiO2 as a nucleus which acts as a silver support [15]. SiO2 is a paramount material for immobilizing nanoparticles on its surface due to its high chemical and thermal stability, chemical inertness, large surface area, and good compatibility with other materials [22,23].  Le et al. [20] evaluated incorporating SiO2-Ag hybrid nanofillers at concentrations of 0.5 to 4.0% wt on water-based acrylic coating. The authors studied the effect of hybrid nanofillers on the abrasion resistance, thermal stability, and antibacterial activity against Escherichia coli. According to the results, incorporating 2.0 % of SiO2-Ag nanofillers simultaneously improves the coating’s mechanical properties, thermal resistance, and antibacterial activity. However, a higher amount of the nanofiller in the matrix (4.0 % wt) showed particle agglomeration, as shown by scanning electron microscopy images, reducing the adhesion and abrasion resistance.

As shown, SiO2 nanoparticles have great potential to be used as carriers and cores for antimicrobial agents, such as AgNPs. However, many works have indicated the tendency for agglomeration and reduction of compatibility of these nanomaterials with organic matrices, especially when the amount of nanofiller is superior to 2.0 wt%. The modification of the SiO2-AgNPs nanofiller surfaces is an alternative to improve their interaction and dispersion in the polymeric matrix. γ–methacryloxypropyltrimethoxysilane (MPS) is an important silane which is widely used as a functionalization agent to promote interfacial interactions between metal oxide nanoparticles and a polymeric matrix, such as dental acrylic resins. The silanization mechanism occurs by M− O− Si (M = metal) bonds between MeO-NPs and silane coupling agents [24].

Several works have shown improvements in silanization of nanofillers used in acrylic-based matrices for dental applications.  Menezes et al. [24] modified the surface of silver vanadate nanorods (AgVO3) using MPS to enhance the dispersion and interaction of this filler with polymethyl methacrylate (PMMA). The authors showed that nanocomposites produced with MPS-modified AgVO3 had higher Shore-D hardness and impact strength values than a nanocomposite with pristine AgVO3 due to the improved dispersion and interaction.

This work reports a pioneer silver-coated SiO2 nanoparticle surface modification through silanization reaction using MPS and their incorporation into dental acrylic matrix. Glaze and soft reliner were used as the organic matrix due to their precise indication to reduce inflammation and distribute a balanced load on the oral tissues during dental repairs. A complete characterization of the MPS-silanized silver-coated SiO2 nanoparticles was performed. Then, an unmodified and modified nanofiller were incorporated into glaze and soft reliner polymers. The effect of the nanofillers in the matrix and antifungal activity was evaluated. Therefore, the aim of this study was to synthesize SiO2-Ag MPS nanoparticles, to characterize them, to test the minimum inhibitory concentration against Candida albicans, and to perform the anti-biofilm test against Candida albicans.”

3 - The Materials and methods section is inappropriately presented as figure 1.

Thank you for the observation. As suggested, we replaced Figure 1 for Tables 1 and 2. Also, the table of “materials used” was removed, and its information was included in the text.

4 - The authors also claim that they obtained SiO2 coated with Ag NPs. They must prove that.- lines 98-105

Thank you for the comments. The results presented in Figures 1 and 2, and in Table 3 showed the deposition of AgNPs on SiO2 surfaces.

Figure 1 presents the FE-SEM images of the SiO2 and SiO2-Ag nanomaterials. SiO2 is constituted of smooth and homogeneous spherical particles. After the silver coating process, new structures are observed on the SiO2 surfaces. Those structures have brighter color when compared to SiO2, indicating to be a heavier element.

A compositional analysis was also performed using SEM-EDX, which showed an incorporation of 7.5 at% of silver after the recovering process. In addition, XRD results showed silver-related peaks after the coating process. Those results proved the production of the desired SiO2-Ag nanoparticles.

The highlighted sentences were included in the results section to prove the silver covering of SiO2 nanoparticles:

SEM: “Figure 2 (a-c) presents the images obtained for SiO2, SiO2-Ag, and SiO2-Ag-MPS. Figure 2.a shows spherical and uniform SiO2 nanoparticles with a smooth and homogeneous surface, and regular size. The average diameter ranged between 468 and 504 nm. After the silver coating process, smaller silver particles can be observed on SiO2 surfaces (Figure 2.b). Those smaller particles (light gray) present a brighter color when compared to SiO2 matrix (dark gray), which corroborates the deposition of AgNPs, considering that heavier elements are brighter in the grayscale of SEM images [28].

SEM-EDX: After the silver-coating, SiO2-Ag showed a significant amount of Ag in the sample composition, over 33.2 wt%, indicating the incorporation of AgNPs on the SiO2 surfaces.

XRD: “The presence of five peaks for the SiO2-Ag samples corresponding to metallic and cubic silver can be observed, corroborating the incorporation of AgNPs on SiO2 surfaces, as shown by Figure 1 and Table 1. The peaks were assigned to the following diffraction planes by JCPDS sheet 01-087-0597: 2θ = 38.11° (111), 2θ = 44.29° (200), 2θ = 64.44° (220), 2θ = 77.39° (311), and 2θ = 81.54° (222).”

5 - The section 2.6 is ambiguous presented.

Thank you for the comments. The section 2.6 describes the nanocomposites fabrication, for glaze and soft reline. It was divided in six different groups, three for the glaze and three for the soft reline (n=12). The highlighted sentence describes them:

“Thus, three different experimental groups were defined for the glaze (Table 1) and three for the soft reline (Table 2) (n = 12), with a negative control group (no nanoparticles), and two experimental groups of nanoparticles, varying the concentration concerning the polymer mass (1.0 % and 2.5%).

6 - The section 3.1.4. must be improved. There are some observations made by assumptions at lines 281-283, 286-288, 292-294. Also the information presented in Figure 3 are a combination between SEM, XRD and FTIR data and must be changed.

Thank you for the comments. Regarding the lines 281-283 and 286-288: The correlation between the bands at 1328 and 1383 cm-1 with the N=O bond, and the reduction of the band at 955 cm-1 related to Si-O-Ag+ formation, were based on highly cited papers, that were included in the sentence:

These bands were related to the symmetrical stretching of the N=O bond of the NO2 ion which could be from AgNO3 used in the coating process [32,34,35].”

Such phenomena may be related to the adhesion of silver to the SiO2 surface. During the coating process, the Si-O-Si and Si-OH bonds broke to form Si-O-Agδ+ [36]. As a result, the intensity of the band related to Si-O-Si and Si-OH bonds is reduced when compared to pristine SiO2 [32].”

Regarding the lines 292-294: The correlation of the bands was based on highly cited papers that also worked with the silanization treatments of nanomaterials. One of the cited papers (REF 39 – Short natural fibre reinforced polyethylene and natural rubber composites: Effect of silane coupling agents and fibres loading), that was used to make the observations about the bands, has been cited over 500 times.

As suggested, Figure 3 were changed into Tables 3 and 4.

7 - Figure 4 does not support the comments, must be replaced or improved.

Thank you for the suggestion. We improved the figure 4. In Figure 4 (a) it is displayed the Petri dish for the minimum inhibitory concentration of SiO2-Ag against Candida albicans. In (b) there is a Petri dish for the minimum inhibitory concentration of SiO2-Ag MPS against Candida albicans.

8 - At figure 5, a cross-section form each material must be added, not only at the surface.

Thank you for the suggestion but there was no time to conduct such evaluation, further studies will perform the analysis.

9 - Section 3.3.2 - add the contact angle values for al samples and then discuss them.

Thank you for the comments. The contact angle for all samples is presented in Tables 5 and 6. We included the following highlighted sentences in the discussion to support the results:”

“The results of the present study agree with the findings of Ziabka et al. [59], in which the contact angle had a statistically significant increase after incorporating the Ag nanoparticles.

10 - The section 3.4. is unclear presented, some information is missing.

Thank you for the comments. The information of corrected as it can be seen below:

“Table 7 showed the CFU/specimen values for the CG, G1, and G2 (p-value not adjusted for ties: 0.245 / p-value adjusted for ties: 0.244). There was no statistical difference for the presence of nanoparticles, regardless of their concentration, given by the Kruskal-Wallis test.

Groups

N

Median

Ave rank

Z-value

CG

12

1300000

20.7

0.87

G1

12

470000

14.3

-1.68

G2

12

1150000

20.5

0.81

Overall

36

18.5

Table 7: CFU/specimen values for the glaze.

Table 8 shows the CFU/specimen values for the CR, R1, and R2 groups (p-value not adjusted for ties: 0.265 / p-value adjusted for ties: 0.264). There was no statistical difference for the presence of nanoparticles, regardless of their concentration, given by the Kruskal-Wallis test. CR presented the lowest CFU/specimen value, so the nanoparticles did not generate benefits in terms of fungal reduction for the soft reliner.

Groups

N

Median

Ave rank

Z-value

CR

12

470000

17.0

-0.59

R1

12

475000

16.0

-1.02

R2

12

1150000

22.5

1.61

Overall

36

18.5

Table 8: CFU/specimen values for the reliner.

11 - In section 4 there are also some observations made by assumptions at lines 369-370, 382-384, 396-397, 402-403, 406-408, 410-414, 424-426, 475-483 and also in Conclusion section- it is not clear the aim of this paper and the results do not support them.

Thank you for the observations. We performed the following changes in the discussion to improve the quality of the observations:

Lines 369-370: We removed the first paragraph of the discussion section, since it was not appropriate to initiate the results discussion.

Lines 382-384: We included more information regarding the AgNPs toxicity:

“Despite its antimicrobial effect, AgNPs can also be cytotoxic to the human body depending on their size, tissue allocation, cellular absorption, surface electric charge, and infiltration competence [42–45]. Several studies have shown that AgNPs cytotoxicity exhibit a dose and time-dependence, being reduced by surface coating techniques [46].”

Lines 396-397, 402-403, and 406-408: We removed some sentences and corrected the paragraph to be more appropriate:

“Thus, the present study aimed to synthesize stable SiO2 nanoparticles to act as carrying material to perform the controlled release of Ag+. After characterization of the material by FE-SEM, it was possible to observe SiO2 nanoparticles that ranged in size from 418 to 502 nm with clustered AgNPs with diameter in the range of 7 to 25 nm. Even though the nanoparticle sizes were adequate in the present study, synthesis methodology improvements must reduce the nanoparticle size. As shown by Devi et al., silica/silver core-shell nanoparticles can act as a antimicrobial agent due to the  SiO2 high surface area, allowing the loading of large amounts of antimicrobial materials and the slow release of bactericidal agents for an extended period [21].

Lines 410-414: We corrected the sentence to be more appropriate:

The EDX characterization of SiO2, SiO2-Ag, and SiO2-Ag-MPS nanoparticles showed a majority of Si, O, and Ag chemical elements. Variation in the mass or atomic percentage of each element was related to the characteristic of each sample. As expected, Si and O elements were the major components for the SiO2 sample. A large amount of silver was observed after the silver coating, indicating its loading on the SiO2 surfaces. The silanization process also made some changes in the EDX results, increasing the number of Si element when compared to the SiO2-Ag sample, which is related to the nanoparticle functionalization.

Lines 424-426: The correlation between the bands at 1328 and 1383 cm-1 with the N=O bond, were based on highly cited papers, that were included in the results sentence:

“These bands were related to the symmetrical stretching of the N=O bond of the NO2 ion which could be from AgNO3 used in the coating process [32,34,35].

Lines 475-483: We corrected the paragraph to be more appropriate:

“The results found in this study promote further understanding of SiO2-Ag silanization mechanism using MPS organosilane and its effects on antifungal activity of acrylic based dental matrices. The results suggested that the sinalization using MPS can produce nanocomposites with superior antifungal activity against C. albicans. The release of silver from the acrylic matrix, and consequently antifungal action, can be improved by developing new studies regarding the controlled release of AgNPs by SiO2 surfaces. Moreover, interference of the functionalization process in the SiO2-Ag hybrid nanofiller must be studied more. Although the functionalization improved the interaction and dispersion of the nanofillers with the polymeric matrix, the effect of this functionalization on the release of AgNPs must be investigated. In addition, it is crucial to carry out further studies to improve the color of the material, evaluate the cytotoxicity, and mechanical properties of the produced materials.”

Reviewer 3 Report

Comments

Summary

This article describes a potential way of synthesis and application of SiO2-Ag NPs incorporated composite, as a soft reline and glaze materials, which are mainly used over full or partial dental prosthesis to prevent excessive pressure on the supporting tissues. Fungal and bacterial infections are very common in surgical wounds and during the use of any dental prosthetics. Considering this aspect, the idea to prepare antimicrobial/ antifungal composite materials for the applications in soft reline and glaze materials are important. This manuscript can be accepted in “Polymers” after a major revision. My comments are summarized below-

Major Comments

1.       Authors should include relevant references in the introduction section to highlight the novelty of their work compared to others. There are some literature available where Ag NPs, ZnO NPs have been used for the development of soft reline materials. I am mentioning one of the references below-

·         Chladek, G., Mertas, A., Barszczewska-Rybarek, I., Nalewajek, T., Żmudzki, J., Król, W., & Łukaszczyk, J. (2011). Antifungal activity of denture soft lining material modified by silver nanoparticles—a pilot study. International Journal of Molecular Sciences, 12(7), 4735-4744.

2.       Authors should explain in detail, what are the significance of using each components including SiO2, Ag NPs, and MPS in their formulated soft reline and glazing materials.

3.       If Authors have only used Ag NPs instead of the SiO2-Ag NPs inorganic-inorganic composite nanoparticles, what will be the difference in the properties?

4.       ZnO, TiO2 NPs have strong antimicrobial activities and these are also used in the formulation of dental prosthetics such as Glass-ionomer cement (GIC). Most importantly, these nanoparticles will not provide too much coloration to the product as like Ag NPs. Do Authors have any thoughts to include these nanoparticles in their formulation? / Why Authors have not considered these two NPs instead of fabricating SiO2@Ag NPs (complicated structure)?

5.       As SiO2 NPs could impart significant effect over the mechanical (rheological properties) and thermal properties of the composite, Author should include the data related to the mechanical and thermal properties changes of the glazing and reline material after the incorporation of SiO2-Ag NPs.

6.       Figure 4 caption should be revised to make it more understandable. From the figure, it is very unclear what are Figure 4a (1 &2) and Figure 4b (1,2, & 3).

7.       What is the shelf-life of the fabricated antifungal glazing and soft reline materials? How long the antifungal activity of the designed composite will sustain?

8.       What will be the fate of MPS after incorporation into the glazing or soft reline material? Is this will be cured during the prosthetic fabrication process or this will be in the monomeric state? Because in the monomeric form MPS is cytotoxic. Clarify this point.

Minor Comments

1.       Line No 16 – Start the sentence with “The objective of the work was……”. Remove “Then”. This sounds odd.

2.       Line No 23 - replace “scanning electronic microscopy” with “scanning electron microscopy”.

3.        Insert Table in Figure 1, Figure 3, and Figure 7 instead of the images and named those as Table 1 and Table 2, and Table 3 respectively. Images are not clear and this representation is not scientifically sound. Change the writing accordingly.

4.       Writing Font and Sizes are not uniform throughout the manuscript (Line No 359 and Conclusion section).

5.       Explanation for Figure 2(2) and Figure 2(3) should come before Figure 3. Change the writing accordingly.

Author Response

Reviewer #3:

This article describes a potential way of synthesis and application of SiO2-Ag NPs incorporated composite, as a soft reline and glaze materials, which are mainly used over full or partial dental prosthesis to prevent excessive pressure on the supporting tissues. Fungal and bacterial infections are very common in surgical wounds and during the use of any dental prosthetics. Considering this aspect, the idea to prepare antimicrobial/ antifungal composite materials for the applications in soft reline and glaze materials are important. This manuscript can be accepted in “Polymers” after a major revision. My comments are summarized below-

Major Comments

1 - Authors should include relevant references in the introduction section to highlight the novelty of their work compared to others. There are some literatures available where Ag NPs, ZnO NPs have been used for the development of soft reline materials. I am mentioning one of the references below-

Chladek, G., Mertas, A., Barszczewska-Rybarek, I., Nalewajek, T., Żmudzki, J., Król, W., & Łukaszczyk, J. (2011). Antifungal activity of denture soft lining material modified by silver nanoparticles—a pilot study. International Journal of Molecular Sciences, 12(7), 4735-4744.

Thank you for the suggestion. We added more relevant references in the introduction section to highlight the importance and novelty of this work, including the suggested reference of Chladek et al. The following highlighted text and references were included:

There is no material available on the dental market which can inhibit microorganisms aggregating on the surface of acrylic resin. Thus, there is a need to create a new material with antimicrobial and antifungal properties whose objective is to form a protective layer capable of inhibiting bacterial and fungal proliferation on the surface of prostheses. In this context, several nanomaterials have been studied by combining biocompatible polymers and inorganic materials, such as SiO2, TiO2, ZnO, and silver, among others [9–13].

Silver nanoparticles (AgNPs) stand out as one of the most intensively investigated systems. They have unique properties, such as excellent electrical conductivity, catalytic activity, non-linear optical behavior, high stability and resistance to oxidation process, and antimicrobial effects, making them potential candidates for applications in different areas including sensors, textiles, electronics, catalysis, food packaging, and medical therapy [14–16]. The use of AgNPs expanded in the dental field because of their bactericidal and bacteriostatic activity against microorganisms such as fungi, bacteria, and viruses [17,18]. Chladek et al. [13] modified silicon soft line polymer with AgNPs (10 to 200 ppm of AgNPs) and evaluated the antifungal efficacy (AFE) of the developed nanocomposites. The results showed AFE varying between 16.4% (10 ppm) to 52.2 % (200 pm) for the examined samples. According to the authors, increasing the amount of AgNPs increased the antifungal behavior.

However, the work of Habibzadeh et al. [19] showed that the addition of AgNPs to silicone soft liner reduced its tensile bond strength. The authors also reported that the release of AgNPs into the oral cavity may increase its toxicity, needing a controlled release to maintain the optimal balance between the biocompatibility and antifungal activity of the nanocomposite. The cited works showed the potential of AgNPs to be used as antibacterial and antifungal materials; however, further studies should be carried out in order to control AgNP toxicity and its effects on polymeric mechanical properties.

In this context, the use of SiO2 nanoparticles have been showing great potential to be used as carrying material for antimicrobial particles. Metal oxides are commonly used due to their stability, biocompatibility, and for being essential minerals for human health [9]. Gad et al. [10] incorporated SiO2 nanoparticles into a soft relining denture material to reduce Candida albicans adhesion. According to the results, the addition of 0.25 and 0.5 % of nano-SiO2 significantly reduced adhesion of C. albicans fungus into acrylic soft liner. However, incorporating 1.0 % of nano-SiO2 resulted in higher C. albicans counts when compared to the 0.25 and 0.5 % samples. The authors suggested that higher nano-SiO2 concentration can lead to their agglomeration in the polymeric matrix.

The combination of antifungal and antibacterial properties of AgNPs with the biocompatibility and stability of SiO2 nanoparticles has great potential. Loading AgNPs in SiO2 surfaces has showed great potential to regulate the solubility and toxicity of Ag+ free ions by controlling their release [20]. SiO2 nanoparticles enable loading large amounts of antimicrobial materials and the slow release of bactericidal agents for a prolonged period, being potentially used as carriers and a core for antimicrobial agents [21]. Many works have been developing techniques to use silver as nanomaterials for biomedical applications by improving its dispersibility, stability, and prolonging the release time. El-Nour et al. studied a methodology to control the release of Ag+ ions by incorporating SiO2 as a nucleus which acts as a silver support [15]. SiO2 is a paramount material for immobilizing nanoparticles on its surface due to its high chemical and thermal stability, chemical inertness, large surface area, and good compatibility with other materials [22,23].  Le et al. [20] evaluated incorporating SiO2-Ag hybrid nanofillers at concentrations of 0.5 to 4.0% wt on water-based acrylic coating. The authors studied the effect of hybrid nanofillers on the abrasion resistance, thermal stability, and antibacterial activity against Escherichia coli. According to the results, incorporating 2.0 % of SiO2-Ag nanofillers simultaneously improves the coating’s mechanical properties, thermal resistance, and antibacterial activity. However, a higher amount of the nanofiller in the matrix (4.0 % wt) showed particle agglomeration, as shown by scanning electron microscopy images, reducing the adhesion and abrasion resistance.

As shown, SiO2 nanoparticles have great potential to be used as carriers and cores for antimicrobial agents, such as AgNPs. However, many works have indicated the tendency for agglomeration and reduction of compatibility of these nanomaterials with organic matrices, especially when the amount of nanofiller is superior to 2.0 wt%. The modification of the SiO2-AgNPs nanofiller surfaces is an alternative to improve their interaction and dispersion in the polymeric matrix. γ–methacryloxypropyltrimethoxysilane (MPS) is an important silane which is widely used as a functionalization agent to promote interfacial interactions between metal oxide nanoparticles and a polymeric matrix, such as dental acrylic resins. The silanization mechanism occurs by M− O− Si (M = metal) bonds between MeO-NPs and silane coupling agents [24].

Several works have shown improvements in silanization of nanofillers used in acrylic-based matrices for dental applications.  Menezes et al. [24] modified the surface of silver vanadate nanorods (AgVO3) using MPS to enhance the dispersion and interaction of this filler with polymethyl methacrylate (PMMA). The authors showed that nanocomposites produced with MPS-modified AgVO3 had higher Shore-D hardness and impact strength values than a nanocomposite with pristine AgVO3 due to the improved dispersion and interaction.

This work reports a pioneer silver-coated SiO2 nanoparticle surface modification through silanization reaction using MPS and their incorporation into dental acrylic matrix. Glaze and soft reliner were used as the organic matrix due to their precise indication to reduce inflammation and distribute a balanced load on the oral tissues during dental repairs. A complete characterization of the MPS-silanized silver-coated SiO2 nanoparticles was performed. Then, an unmodified and modified nanofiller were incorporated into glaze and soft reliner polymers. The effect of the nanofillers in the matrix and antifungal activity was evaluated. Therefore, the aim of this study was to synthesize SiO2-Ag MPS nanoparticles, to characterize them, to test the minimum inhibitory concentration against Candida albicans, and to perform the anti-biofilm test against Candida albicans.”

2 - Authors should explain in detail, what are the significance of using each components including SiO2, Ag NPs, and MPS in their formulated soft reline and glazing materials.

Thank you for the suggestion. We included in the introduction the importance of each component of the nanocomposite:

SiO2: In this context, the use of SiO2 nanoparticles have been showing great potential to be used as carrying material for antimicrobial particles. Metal oxides are commonly used due to their stability, biocompatibility, and for being essential minerals for human health [9].

SiO2 nanoparticles enable loading large amounts of antimicrobial materials and the slow release of bactericidal agents for a prolonged period, being potentially used as carriers and a core for antimicrobial agents [21].”

AgNPs: Silver nanoparticles (AgNPs) stand out as one of the most intensively investigated systems. They have unique properties, such as excellent electrical conductivity, catalytic activity, non-linear optical behavior, high stability and resistance to oxidation process, and antimicrobial effects, making them potential candidates for applications in different areas including sensors, textiles, electronics, catalysis, food packaging, and medical therapy [14–16]. The use of AgNPs expanded in the dental field because of their bactericidal and bacteriostatic activity against microorganisms such as fungi, bacteria, and viruses [17,18].”

Combination of SiO2 and AgNPs: The combination of antifungal and antibacterial properties of AgNPs with the biocompatibility and stability of SiO2 nanoparticles has great potential. Loading AgNPs in SiO2 surfaces has showed great potential to regulate the solubility and toxicity of Ag+ free ions by controlling their release [20].

Many works have been developing techniques to use silver as nanomaterials for biomedical applications by improving its dispersibility, stability, and prolonging the release time.”

MPS: As shown, SiO2 nanoparticles have great potential to be used as carriers and cores for antimicrobial agents, such as AgNPs. However, many works have indicated the tendency for agglomeration and reduction of compatibility of these nanomaterials with organic matrices, especially when the amount of nanofiller is superior to 2.0 wt%. The modification of the SiO2-AgNPs nanofiller surfaces is an alternative to improve their interaction and dispersion in the polymeric matrix. γ–methacryloxypropyltrimethoxysilane (MPS) is an important silane which is widely used as a functionalization agent to promote interfacial interactions between metal oxide nanoparticles and a polymeric matrix, such as dental acrylic resins. The silanization mechanism occurs by M− O− Si (M = metal) bonds between MeO-NPs and silane coupling agents [24].”

Soft reline and glaze: Glaze and soft reliner were used as the organic matrix due to their precise indication to reduce inflammation and distribute a balanced load on the oral tissues during dental repairs.

3 - If Authors have only used Ag NPs instead of the SiO2-Ag NPs inorganic-inorganic composite nanoparticles, what will be the difference in the properties?

AgNPs stand out as one of the most used nanomaterial able to improve the antibacterial and antifungal activity of nanocomposites. Their bactericidal and bacteriostatic activity is well known and diffused in the scientific reports. However, the use of AgNPs have been questioned due to their toxicity to the human body. Also, many works have showed that AgNPs can affect the mechanical properties of organic matrices negatively. In this way, the use SiO2 as carrying material for AgNPs has great potential to regulate the toxicity of the Ag+ free ions. Also, some works showed that SiO2 nanoparticles did not interfere in the polymeric mechanical properties.

The antifungal activity should be ever higher if the AgNPs were used instead of the SiO2-AgNPs. However, the produced nanocomposite will not be appropriated for practical application due to the risk of toxicity associated with the Ag+ uncontrolled release. So, the use of SiO2-Ag combines the antifungal and antibacterial properties of AgNPs with the biocompatibility and stability of SiO2 nanoparticles.

The following highlighted sentence was included in the introduction to show some AgNPs disadvantages:

The cited works showed the potential of AgNPs to be used as antibacterial and antifungal materials; however, further studies should be carried out in order to control AgNP toxicity and its effects on polymeric mechanical properties.

Also, the following highlighted sentence was included in the introduction to show the benefits of SiO2 and AgNPs combination:

The combination of antifungal and antibacterial properties of AgNPs with the biocompatibility and stability of SiO2 nanoparticles has great potential. Loading AgNPs in SiO2 surfaces has showed great potential to regulate the solubility and toxicity of Ag+ free ions by controlling their release [20].”

      Similar information was presented in the results discussion:

Despite its antimicrobial effect, AgNPs can also be cytotoxic to the human body depending on their size, tissue allocation, cellular absorption, surface electric charge, and infiltration competence [42–45]. Several studies have shown that AgNPs cytotoxicity exhibit a dose and time-dependence, being reduced by surface coating techniques [46]. At low concentrations, AgNPs have several applications in different areas of dentistry, such as endodontics, dental prosthesis, implantology, and restorative dentistry. Their incorporation aims to prevent or reduce bacterial and fungal colonization in dental materials, improving patient’s oral health and quality of life [47]. Cytotoxicity was not evaluated in this study, as the nanoparticle had a SiO2 core, being described in other studies as a stabilizer of antimicrobial agents, reducing the cytotoxicity of the materials [21,48].”

4 - ZnO, TiO2 NPs have strong antimicrobial activities and these are also used in the formulation of dental prosthetics such as Glass-ionomer cement (GIC). Most importantly, these nanoparticles will not provide too much coloration to the product as like Ag NPs. Do Authors have any thoughts to include these nanoparticles in their formulation? / Why Authors have not considered these two NPs instead of fabricating SiO2@Ag NPs (complicated structure)?

Thank you for your comment. New studies will be conducted with the use of these two NPs. For this study, we choose the SiO2-Ag NPs because of the following sentence added to the article:

“Silver nanoparticles were selected because of the previously reported antifungal effect [62,63]. SiO2 was the vehicle that carried the silver nanoparticles, aiming to reduce its toxicity and to regulate its release, which could extend the antimicrobial effect [21,48].”

5 - As SiO2 NPs could impart significant effect over the mechanical (rheological properties) and thermal properties of the composite, Author should include the data related to the mechanical and thermal properties changes of the glazing and reline material after the incorporation of SiO2-Ag NPs.

Thank you for the comment. We agree with the reviewer that the mechanical and thermal experiments are very important to support real clinical application.

Several works in the literature showed an improvement of the mechanical and thermal properties of polymeric nanocomposites after the incorporation of SiO2 nanoparticles. To perform the same studies for the SiO2-Ag hybrid nanofiller will be essential. However, we detected that the critical gap in the literature regarding the use of soft reliner and glaze is adhesion of biofilm colonies, resulting in oral lesions as prosthetics stomatitis. For this reason, the first step of our research was to develop a route for SiO2-Ag synthesis and silanization with MPS, and the proper addition of these nanofillers in acrylic dental matrices. At this point, we observed that the process improved the dispersion of the nanofillers and also the antifungal activity. These very promising results motivate the development of further studies on the mechanical and thermal properties of the material, that are under development at our laboratory.

6 - Figure 4 caption should be revised to make it more understandable. From the figure, it is very unclear what are Figure 4a (1 &2) and Figure 4b (1,2, & 3).

Thank you for the comment. We agree with the reviewer and changed the figure.

7 - What is the shelf-life of the fabricated antifungal glazing and soft reline materials? How long the antifungal activity of the designed composite will sustain?

            This is the objective of new studies that will be performed soon. As for the glaze and the soft reline, its clinical application is of 6 months length, according to the manufacturer’s instructions.

8 - What will be the fate of MPS after incorporation into the glazing or soft reline material? Is this will be cured during the prosthetic fabrication process or this will be in the monomeric state? Because in the monomeric form MPS is cytotoxic. Clarify this point.

Thank you for the observation. Indeed, some literatures cited MPS monomer as cytotoxic. However, due to the similarity between the MPS and dental materials structures (ethyl methacrylate and methyl methacrylate), the C=C bonds in MPS methacrylate groups can copolymerize with the C=C bonds of the ethyl methacrylate and methyl methacrylate. In this way, MPS toxicity is reduced. In addition, even if some residual MPS monomer still remains in the composition, the amount of MPS used is so low that will not interfere in the dental material toxicity.

The highlighted sentence was included in the discussion to clarify this point:

“The MPS organosilane was used as coupling agent due to its ability to couple with organic matrix, such as acrylic-based polymers, and to inorganic fillers, like the SiO2-Ag hybrid nanoparticles [51]. The inserted functional groups, especially the methacryloxy group, are expected to improve both the dispersion and interaction of the nanofillers in the acrylic matrix. According to the manufacturer, the soft reliner is constituted of ethyl methacrylate, while the glaze is constituted of methyl methacrylate. Thus, the methacrylate group is present in soft reliner, glaze, and MPS structure and this molecular structure similarity favors the interaction between the components [52]. In addition, the C=C bond in the MPS methacrylate group can copolymerize with the C=C bond of ethyl methacrylate (soft reliner) and methyl methacrylate (glaze) during the polymerization reactions of the dental materials [51].

The toxicity of MPS monomer in living tissues remains uncertain due to the lack of studies in the area [53,54]. However, in the present work, MPS is used as functionalization agent for SiO2-Ag nanoparticles, being used in a very low concentration. In addition, as cited, MPS methacrylate groups are copolymerized with the soft reliner and glaze monomers, reducing its toxicity.”

Minor Comments

1 - Line No 16 – Start the sentence with “The objective of the work was……”. Remove “Then”. This sounds odd.

Thank you for the observation. As suggested, we removed the word “Then” in the sentence.

2 - Line No 23 - replace “scanning electronic microscopy” with “scanning electron microscopy”.

Thank you for the observation. As suggested, we changed “scanning electronic microscopy” by “scanning electron microscopy”.

3 - Insert Table in Figure 1, Figure 3, and Figure 7 instead of the images and named those as Table 1 and Table 2, and Table 3 respectively. Images are not clear and this representation is not scientifically sound. Change the writing accordingly.

Thank you for the observation. As suggested, we replaced the figures 1, 3, and 7 for the tables 1 to 8. Also, the table of “materials used” was removed, and its information was included in the text.

4 - Writing Font and Sizes are not uniform throughout the manuscript (Line No 359 and Conclusion section).

Thank you for the observation. The Font and Sizes were standardized in the text.

5 - Explanation for Figure 2(2) and Figure 2(3) should come before Figure 3. Change the writing accordingly.

Thank you for the suggestion. We divided Figure 2 into Figures 1, 2, and 3, and added the explanations before the figure, as suggested.

Round 2

Reviewer 1 Report

Accept in present form

Reviewer 2 Report

The revised manuscript is a clearly improved version compared with the one initial sent. The authors clearly present the state-of-the-art in the field, the purpose of these materials specifying the original part of the work. Also, authors added additional analyses to characterize the nanoparticles and their composites and improved quality figures in the text.

I recommend the publication of this version in the Polymer journal.

Reviewer 3 Report

Congratulations! This manuscript can be accepted in its present form.